# Topographic organization of eye-position dependent gain fields in human visual cortex

Jasper H. Fabius [1,2] ✉, Katarina Moravkova[1] & Alessio Fracasso [1] ✉

The ability to move has introduced animals with the problem of sensory ambiguity: the position of an external stimulus could change over time because the stimulus moved, or because the animal moved its receptors. This ambiguity can be resolved with a change in neural response gain as a function of receptor orientation. Here, we developed an encoding model to capture gain modulation of visual responses in high field (7 T) fMRI data. We characterized population eye-position dependent gain fields (pEGF). The information contained in the pEGFs allowed us to reconstruct eye positions over time across the visual hierarchy. We discovered a systematic distribution of pEGF centers: pEGF centers shift from contra- to ipsilateral following pRF eccentricity. Such a topographical organization suggests that signals beyond pure retinotopy are accessible early in the visual hierarchy, providing the potential to solve sensory ambiguity and optimize sensory processing information for functionally relevant behavior.

Around 600 million years ago animals populating the ocean's floor started moving, leaving fossilized trails that can be observed at the present day. Self-motion brings fundamental evolutionary advantages, as animals that move can find new sources of food or flee from hazards. However, the ability to move comes at a cost: it introduces ambiguity about the source of sensory input. Here, we focus on visual processing and eye movements as a paradigmatic example of sensory ambiguity.

Consider looking at "La nuit étoilée" by van Gogh (Fig. 1A). When watching complex scenes like in the painting, we constantly move our eyes[1]. As we are scanning the town's illuminated skyline, each neuron in our visual cortex responds to only a small portion of the visual field[2]. This portion is their receptive field, which is anchored to the center of gaze. Thus, with each movement of our eyes, the receptive fields are displaced from their former locations. A receptive field that is covering a star at one fixation, might be covering a part of the dark night sky at the next (Fig. 1B).

In visual areas, the arrangement of photoreceptors in the retina is maintained, giving rise to visual field maps, or retinotopy[3]. As the example illustrates, the mere topography of photoreceptors is not sufficient to accurately localize objects in space because eye movements create sensory ambiguity. How does the visual system solve this ambiguity? One hypothesis is that it encodes information about the orientation of the eyes and combines it with the retinotopic location of the visual input, implemented in the response gain of visual neurons: identical retinotopic stimulation leads to different response strengths when eye orientations are different[4] (Fig. 1D). The strength of gain modulations for different eye positions can be described by an eye-position dependent gain field (EGF; Fig. 1C). Computational work has demonstrated that such gain modulation of visual responses by eye position allows for accurate localization of sensory events beyond their retinotopic location, transforming purely retinotopic coordinates into eye-position invariant coordinates[5–7].

Evidence for EGFs has been found with neurophysiological recordings in primate areas 7a[4,8,9], LIP[8,10], V6[11], V3A[12], MT/MST[10,13], V1[14–18], and V4[14]. In humans, similar gain modulations of visual responses have been observed with functional magnetic resonance imaging (fMRI) when observers fixated at different positions for a prolonged time while neural responses to visual stimuli were measured[19–21]. Moreover, electrophysiological recordings have started to shed light on aspects of the organizational principles of EGFs[12,17]. However, our understanding of EGF properties at the population level has been partially restricted by the scope of the recording technique (e.g., electrophysiological recordings of a few hundred neurons) or the experimental paradigm (e.g., examining

[1]School of Psychology and Neuroscience, University of Glasgow, Glasgow, UK. [2]OnePlanet Research Center, Imec, Wageningen, The Netherlands.
✉e-mail: jasper.fabius@glasgow.ac.uk; alessio.fracasso@glasgow.ac.uk

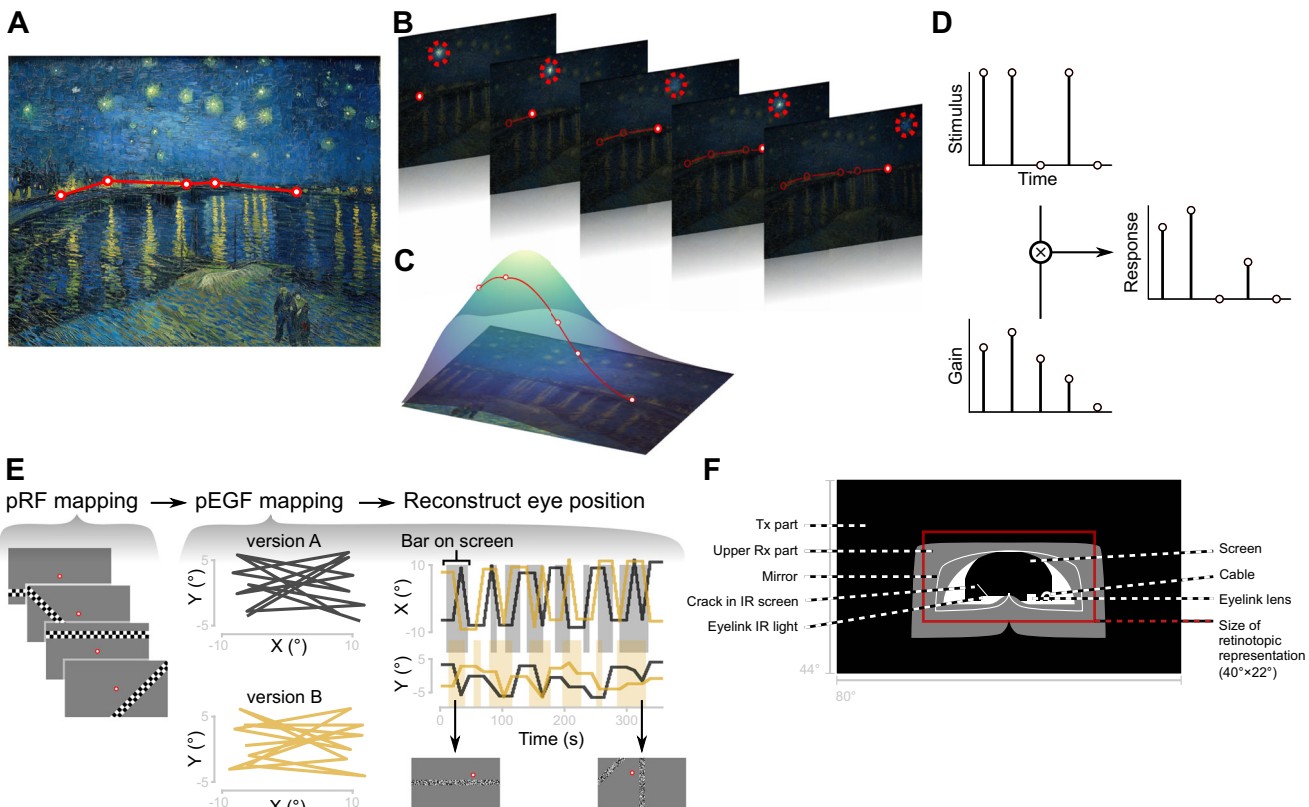

**Fig. 1 | Conceptualization of eye-position dependent gain fields and methods.**
**A** Series of fixations (red points and lines) made across "La nuit étoilée" (van Gogh, 1888). **B** Example receptive field to the upper right of the fixation point. **C** Example gain field. **D** Given the series of fixations in **A** and the receptive field location in **B**, a star from the painting stimulates the receptive field on fixations 1, 2, and 4. The gain modulation, that follows from the gain field in **C**, is strongest on fixation 2. The multiplication of stimulus strength and eye-position dependent gain modulation determines the response strength. **E** In our fMRI experiment, we measured populations of neurons contained within a single voxel. We first mapped population receptive fields (pRF) using a standard moving bar paradigm. Next, we used version A of a novel eye-movement task (black) to map population eye-position dependent gain fields (pEGF): pEGF mapping. While the participants were performing eye movements, high-contrast flickering bars were presented at various times in

different configurations. After estimating each voxel's pRF and pEGF, we reconstructed eye movements using version B of the eye-movement task (yellow). The trajectory of the eyes and the presentation of the bars was uncorrelated between the version A and B. The black lines represent eye positions in version A of the pEGF mapping paradigm, yellow lines represent eye positions in version B. The same color coding is used in the timeseries. In the timeseries, the gray shaded areas represent time windows when high-contrast bars were displayed on the screen in version A; the yellow shaded areas show the same for version B. **F** To capture the visual input in the eye-movement tasks, we created a retinotopic representation that included both the bars and various elements that were present in the peripheral visual field. A movie of the retinotopic representation of each task can be seen in Mov. S1-3. Source data are provided as a Source Data file.

modulations for a limited set of separate and static eye positions). A systematic characterization of gain-field properties allows us to ask questions about the distributional features of EGFs (e.g., are there biases for a particular viewing direction?) and its extension to human parietal cortex. Moreover, it offers the possibility to probe the existence of a potential underlying topographical organization for EGFs. This idea has been originally put forward in the discussion of Andersen and colleagues (1985), and has been a point of debate ever since, with some studies describing hints of topography in some cortical areas[12,17,18] and others observing no systematicities at the population level[8,9,11,13].

Here, we present an encoding model that characterizes visual responses and EGFs at the level of single voxels in human 7 T functional magnetic resonance imaging (fMRI) data. EGFs have been generally studied under static conditions both in human and non-human primates, with participants keeping stable fixation while being presented with a visual stimulus[20–22]. However, we cannot a priori assume that a mechanism studied in a passive setting would transfer in an active setting following the same principles. We introduce a novel approach by probing EGFs in an active setting, with participants moving their eyes along given trajectories while being presented with high contrast visual stimuli.

The model follows a parsimonious approach and is built on first principles: the retinotopic organization of visual cortex and its response to contrast. First, we show that the population receptive field model (pRF)[23] can capture visual responses elicited by eye movements that bring a stimulus into a pRF, capturing contrast-based responses from visual cortex in an active setting. Second, building upon the pRF responses, we estimate population eye-position dependent gain fields (pEGF) as two dimensional gaussians, which we validate by using the pEGFs to reconstruct eye positions from an independent dataset. Third, our encoding model allows for the exploration of previously unknown, large-scale systematicities in the distributions of EGF parameters at the population level. In early visual areas we observed that pEGF centers follow a topographic organization along the eccentricity of pRFs.

## Results

### Population receptive fields capture visual responses induced by eye movements

Before incorporating EGFs into our encoding model, we first needed to establish that the model captures visual responses adequately. To this end, our model starts with the estimation of population receptive fields (pRF)—i.e., voxel-based receptive fields[23]. A pRF can be described

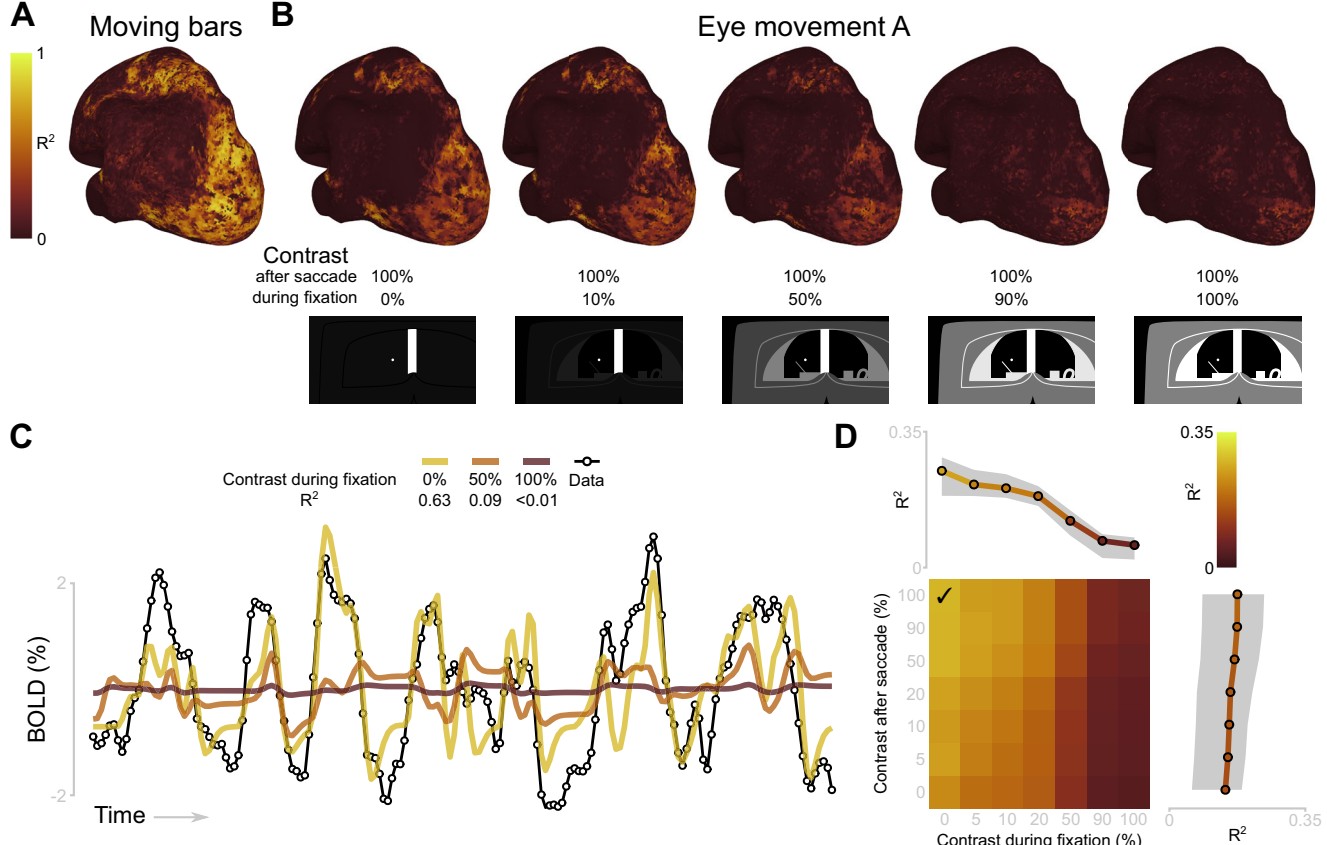

**Fig. 2 | Creating the optimal retinotopic representation of the visual input to capture responses in visual cortex during the eye-movement task. A** Left hemisphere surface of an example participant showing the variance explained ($R^2$) by the pRFs for the moving bar paradigm—these data were used to estimate the pRFs. **B** Same example hemisphere as in A, now overlaid with the variance explained for different configurations of the retinotopic representation of the visual input during the eye-movement task. In all configurations, the contrast of the peripheral stimuli increased to the same level as the vertical bars after a saccade and was reduced to a lower contrast level during periods of fixation (ranging from 0 to 100%). The different contrast levels of the peripheral elements are displayed at the bottom. The white vertical bar is the high-contrast flickering checkboard bar. The point in the center is the fixation point. **C** Example BOLD time series from the

same participant (black). Lines from yellow to dark red represent scaled predicted time series based on this voxel's pRF. For all predictions, the contrast of the peripheral stimuli was increased to 100% after a saccade. The contrast levels during fixation vary according to the legend. **D** Group level average variance explained for all contrast configurations in V1. First, we selected the voxels whose time series were explained best by their pRF, as estimated with the moving bar paradigm (median split). Next, we took the median $R^2$ of those voxels per participant and per contrast configuration. Finally, we computed the group median (color values in bottom left, and lines in top and bottom right) and interquartile range (shaded gray area). The check mark indicates the contrast configuration that was used in the other analyses. Source data are provided as a Source Data file.

by four parameters: center (x, y), standard deviation and an exponent for compressive spatial summation[23,24]. Notably, the pRF locations are expressed in retinotopic coordinates. Neurophysiological recordings have previously demonstrated that visual receptive fields shift along with the direction of gaze, staying anchored to the fovea[25,26]. Moreover, an abrupt change to the input in a neuron's receptive field leads to consistent responses, irrespective of whether the change was passive (i.e., a flashed stimulus) or active (i.e., as the result of a saccade)[27–29]. Thus, we set out to verify whether pRFs can capture active visual responses when eye movements relocate a stimulus into the pRF of a voxel.

We estimated a pRF for each voxel using a standard pRF mapping paradigm. In the scanner, participants were shown a moving bar stimulus as they maintained fixation at the center of the screen (Fig. 1E), while we measured the blood-oxygenation-level-dependent (BOLD) signal. We created 5400 hypothetical pRFs and calculated a corresponding predicted signal given the stimulus that was shown to the participants (Supplementary Movie 1). We then compared each of the predicted time series to the measured BOLD signal. The pRF whose prediction most closely matched the measured signal of a voxel (highest variance explained, $R^2$) was assigned to that voxel. The pRF parameters were used to define the borders between visual areas[3]. As

expected, pRF size scales with eccentricity ($F(1,1047.2) = 835.2$, $p < 0.0001$) and visual area ($F(9,1046.1) = 51.89$, $p < 0.0001$). Here, we combined relatively smaller visual areas into single regions of interest (ROI) with approximately the same number of voxels as the larger areas. We obtained a median variance explained ($R^2$) varying between 0.66 (V2) and 0.40 (IPS2-3-4).

After estimating each voxel's pRF, we tested to what extent the pRF could capture visual responses elicited during an eye-movement task. In this task, visual stimulation was retinotopically similar to the standard moving bar paradigm. In the standard paradigm, a high-contrast bar moves across the screen in small steps. In our eye-movement task, participants were making eye movements across the screen in small steps while high-contrast, flickering bars were presented statically (Fig. 1E). The retinotopic stimulation is comparable: in both tasks, the bars sweep across the retina in small steps. See Supplementary Movie 1 and 2 for an example of the moving bar task and the eye-movement task (version A), respectively.

To predict a voxel's response given its pRF, we created a retinotopic representation of the visual input presented in the scanner as the participants performed the eye-movement task. The retinotopic representation was a downsampled movie (22 × 40° compressed to 149 × 267 pixels, 6 Hz) in which all the visual elements present within

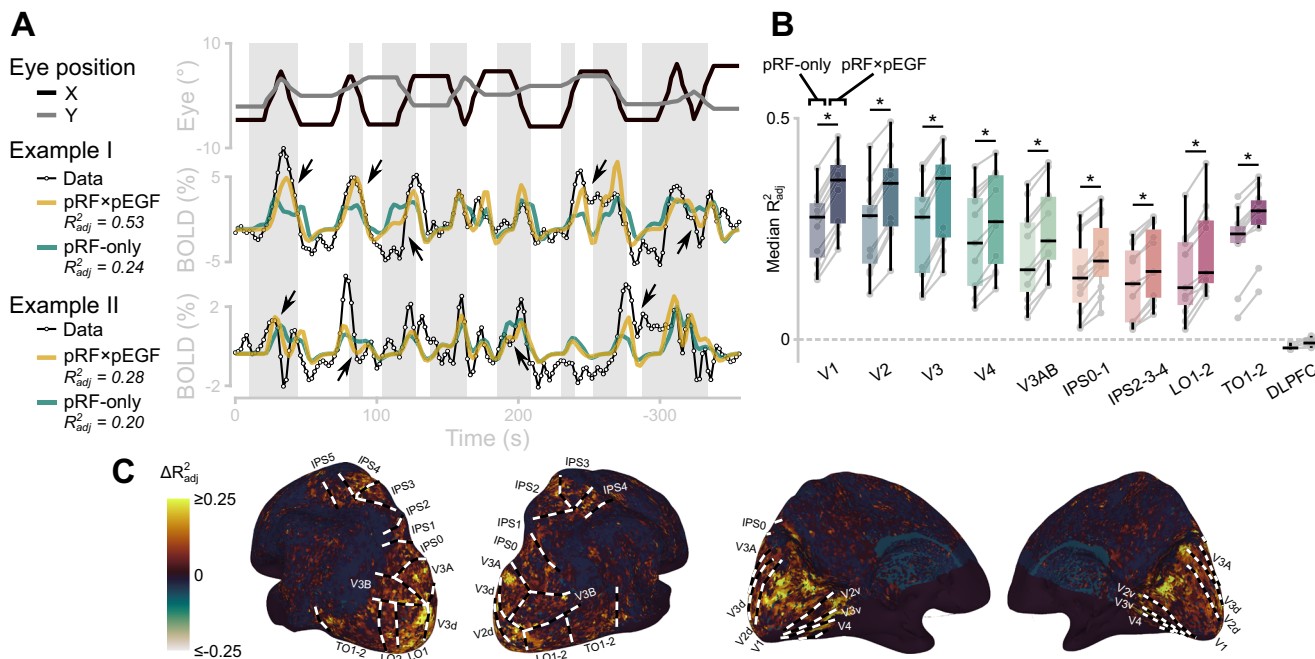

**Fig. 3 | Population eye-position dependent gain fields (pEGF) improve predicted BOLD signal time series based on population receptive fields (pRF).**
**A** Two example BOLD time series from V1. Eye position time series (top black, gray lines) and bar displays (gray rectangles) of version A of the eye-movement task are plotted for reference. Example I, voxel for which the pRF-only prediction (green) markedly improved when the pEGF was added to the pRF (yellow). Arrows indicate parts of the time series where the prediction was most notably improved. Example II, voxel that is representative of the median adjusted $R^2$ in V1, both for the pRF-only and the pRF×pEGF models. **B** Change in adjusted $R^2$ per visual ROI. Per ROI, the light, left boxplot represents the pRF-only model, the right, dark boxplot represents the pRF×pEGF model. Gray lines and points represent single participants ($N = 11$). In the box plot, the center is the median, the box bounds are Q1 and Q3, the

whiskers extend to the highest/lowest values with a max/min of 1.5× the IQR, data beyond these limits are shown as points. Because pRFs were initially fitted on the moving bar data, we computed the median adjusted $R^2$ ($R^2_{adj}$) for this figure from the 50% of most visually responsive voxels in each ROI. The $R^2_{adj}$ adjusts for the number of parameters in a model, as such it can be lower than zero. Horizontal bars with an asterisk indicate that the difference in adjusted $R^2$ is larger than zero (i.e., a difference of zero is outside the Bonferroni-corrected bootstrapped 95%-confidence interval). **C** Surface maps from one example participant of the change in adjusted $R^2$ from the pRF-only model to the pRF × pEGF model. Warm colors indicate an increased fit, cool colors indicate a decreased fit. Black and white lines indicate the manually drawn boundaries between ROIs. Source data are provided as a Source Data file.

the scanner moved with respect to the point where the participant was fixating (Supplementary Movie 2). These elements included the high-contrast bars and the fixation point, as well as various elements in the peripheral visual field, such as the screen edge, coil, and the mirror (Fig. 1F). Because visual responses are evoked when abrupt changes occur in the receptive field[27], we explored how the peripheral elements could be best incorporated into the retinotopic representation of the visual input.

First, we varied the contrast of the peripheral elements from 0 to 100% with respect to the contrast of the bars. There were two moments of contrast increase of these elements: one after a saccade (for 333 ms, i.e., two frames at 6 Hz) and one during fixation (all the other frames). The contrast change was implemented instantaneously, without continuous transition (i.e., contrast incrementally increased between frames), because this would not be an accurate representation of contrast change between two consecutive frames after a saccade.

To capture the saccade latency in the retinotopic representation of the stimulus, the fixation point jumped to its next location before being recentered. The saccade latency was 1 frame at 6 Hz or 166 ms (cf. actual median latency = 135 ms, Supplementary Fig. 1D). Next, we passed the differently configured retinotopic representations of the visual input through the pRFs of each voxel to create a predicted BOLD response for each configuration. We scaled all predictions to the data using ordinary least squares regression and computed the resulting $R^2$ (see Methods - Retinotopic representation of visual input in eye-movement tasks).

Example cortical surface maps of variance explained by different contrast configurations are displayed in Fig. 2B. Qualitatively, these

maps follow the same pattern as the map of variance explained obtained by estimating the pRFs from the moving bar paradigm (see Fig. 2A, B, leftmost surface). Example time series of a single voxel, combined with a set of predictions from three different stimulus configurations are shown in Fig. 2C. We used the optimal configuration of contrasts during fixation and after saccade offset based on the median over all participants' V1 voxels. The optimal contrast of the peripheral elements during fixation was 0%, after a saccade the contrast was increased to 100% of the contrast of the flickering checkerboard bars (Fig. 2D).

After establishing the optimal contrast levels, we examined how long the contrast of the peripheral elements should be increased after each saccade in the retinotopic representation of the visual input. For this, we used a post-saccadic contrast increase of 100% and a contrast of 0% for the periods of fixation. Because we created the retinotopic representations with 6 frames per second, we examined durations of 167, 333, 500, 667, 1000, 1333, 2000, 4000, and 5333 ms. Again, we generated predictions from the combination of each voxel's pRF and the different configurations of the retinotopic representation. For all further analyses, we used the contrast increase duration of 500 ms, which was the average optimum for voxels in early visual cortex (Supplementary Fig. 1F). Please note that the optimization (contrast and time after saccade) of the stimulus and model fitting were performed on task version A (see Fig. 1E). Reconstruction was performed on task version B (see Fig. 1E) without further optimization to test generalization and avoid overfitting.

The optimal configuration of the peripheral elements incorporated into the stimulus model consisted of post-saccadic contrast

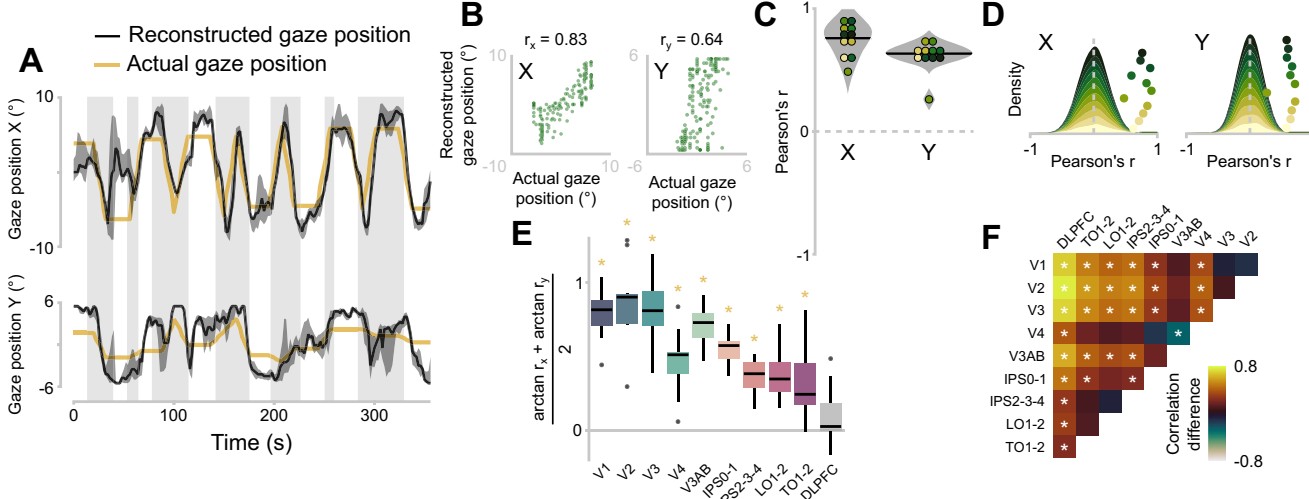

**Fig. 4 | Eye position reconstruction. A** Horizontal (top) and vertical (bottom) components of the median reconstructed eye position (black) and actual target position (yellow) of version B of the eye-movement task. Reconstruction was performed using all visual ROIs. Dark gray shaded area represents the interquartile range. Gray rectangles are periods when the flickering bars were on the screen. **B** Scatter plots of reconstructed eye position against the actual eye position for an example participant. **C** Pearson's correlation coefficients for all participants (points). Violins represent kernel density estimates. Black line is the group median. **D** Stacked kernel density estimates of the correlation coefficients from the permutation ($N = 1000$). Colors represent single participants. Points represent the correlations obtained with the true, estimated pEGF parameters (same is in C). The height of the points is arbitrary but follows the order of the stacking of the kernel density estimates. **E** Correlations between actual eye position and reconstructed positions from single visual ROIs and a control area in the dorsolateral prefrontal cortex (DLPFC). The horizontal and vertical components were combined into a single correlation estimate (the Fisher transformed average of the two components) to compare the ROIs with each other. Correlations per ROI were estimated by bootstrapping a fixed number of voxels of each ROI per participant ($N = 11$ participants). This ensures that differences in reconstruction quality are not due to differences in size of the ROIs. Box plots made with the same conventions as Fig. 3B. Yellow asterisks above the boxplot indicate that the group median is significantly different from zero (i.e., that zero is outside the Bonferroni-corrected bootstrapped 95%-confidence interval). **F** Bootstrapped pairwise comparison of the reconstruction quality between the different ROIs. Red/yellow indicates that the correlation estimate of the ROI in the row is higher than that of the ROI in the column. Colors represent the median difference across the eleven participants. White asterisks indicate a significant difference (alpha = 0.05, Bonferroni corrected for multiple tests, see Supplementary Information – Statistics Output). Source data are provided as a Source Data file.

increases (for 500 ms, from 0 to 100%) and no contrast during periods of fixation. The optimal configuration was set based on the increase in $R^2$ in V1 voxels. This stimulus configuration led to a significant increase in $R^2$ of approximately 0.035 in V1, V2, IPS0-1, IPS2-3-4 and TO1-2, but not in the other areas (Fig. S2; Supplementary Note 1).

To summarize, in line with single cell recordings[27], the static peripheral elements act as a visual stimulus when they are brought into a pRF after a saccade (Fig. 2D). The median variance explained by the pRF model in combination with the optimized stimulus configuration (pRF-only model) varied between 0.30 (V1, V2, V3) and 0.14 (LO1-2) across all visual areas. Thus, pRFs can be used to capture visual responses in active eye-movement tasks when the visual input is modeled adequately.

### Eye-position dependent gain fields increase variance explained of pRF model in an eye-movement task

In the previous section, our encoding model was based only on retinotopic pRFs (pRF-only model). In this section, we examined whether the BOLD signal also reflects the eye-position dependent gain modulation observed in neurophysiological recordings. To do so, we extended the pRF-only model with population eye-position dependent gain fields (pEGF). In this extended model (pRF×pEGF model), the gain of the predicted pRF response is modulated by eye position given a specific pEGF (Fig. 1D).

At the level of single neurons, EGFs have classically been described as planar (i.e., two-dimensional planes)[4]. However, they have also been described as various other shapes[30,31] to account for observed spatial non-monotonic features in EGFs[15,32]. Here, we opted to model pEGF at the level of single voxels with an isotropic two-dimensional gaussian described with four parameters: center (x, y), standard

deviation and amplitude. We chose this option for three reasons. First, it is a bounded function (in contrast to planar or polynomial functions), which is convenient to model changes in the BOLD time series. Second, it can account for spatial non-monotonicities in the EGF. Third, at the same time it can approximate planar shapes, in the case of an eccentric 2D gaussian with a (relatively) wide standard deviation[31]. In our modeling strategy we also constrained the pEGFs to be larger than a voxels' pRF[12].

For each voxel, the gain modulation corresponding to the eye positions in the eye-movement task was extracted from this pEGF. We used the same eye positions for every participant, assuming that they were, on average, accurately following the fixation target (as confirmed by the eye-tracking data collected in the scanner; Supplementary Fig. 1A–E). To create a predicted BOLD-signal time series, the predicted response from the pRF-only model was multiplied with the gain modulation, convolved with the hemodynamic response function (HRF), and finally downsampled to 0.5 Hz. We created 3888 hypothetical pEGFs and obtained the best fitting parameters for each voxel (see Methods - Population eye-position dependent gain fields). Importantly, we did not estimate each voxel's pRF again, instead we used the independent pRF estimates from the moving bar paradigm.

To evaluate the extent to which the pEGF improved the fit, we compared the goodness of fit of the pRF-only model and pRF×pEGF model. For this comparison we used the adjusted $R^2$ ($R^2_{adj}$), which corrects for the number of parameters in each model. We examined the changes in $R^2_{adj}$ in the visual areas and an area in the dorsolateral prefrontal cortex (DLPFC) that served as a non-visual control area, where the pRF-only model did not explain the data well (median $R^2 = 0.01$, $R^2_{adj} = -0.02$). Across all participants and visual areas, the median $R^2_{adj}$ increased from 0.20 to 0.27 after the pEGF was added to

the encoding model. At the time series level, the pEGF improved the pRF model by amplifying and suppressing the prediction at various time points (Fig. 3A).

Across all cortical voxels, the increase in median $R^2_{adj}$ mapped onto the visual areas identified with the moving bar stimulus (Fig. 3C). The change in median $R^2_{adj}$ ($\Delta R^2_{adj}$) was larger in all visual ROIs than in the non-visual control area in the DLPFC, where the $\Delta R^2_{adj}$ was 0.004 and not significantly different from 0. This indicates that the pEGF is only of added value when the pRF already captures the visual responses adequately. In addition, there were substantial differences between visual ROIs in the extent to which the pRF×pEGF model improved the $R^2_{adj}$ (Fig. 3B). These differences were tested using a linear mixed-effects model of the difference in $R^2_{adj}$ between the pRF-only and pRF×pEGF models, per participant and ROI. There was a significant effect of ROI (F(9,90) = 30.77, $p < 0.001$). The $R^2_{adj}$ in the early visual areas (V1, V2, V3) increased significantly more ($\Delta R^2_{adj}$ V1 = 0.059, V2 = 0.053, V3 = 0.055) than in higher visual areas ($\Delta R^2_{adj}$ between 0.025 and 0.039; see Supplementary Note 2).

These findings demonstrate that our encoding model with a pRF and pEGF captures signal fluctuations compatible with an eye-position dependent gain modulation. Moreover, they indicate that the eye-position dependent gain modulation is a ubiquitous feature of responses throughout the human visual system[20], including higher intra-parietal sulcus locations.

### Eye position can be reconstructed continuously from fMRI data

If the pEGF captures a modulation of the visual response that is correlated with eye position, we should be able to reconstruct eye position using the pEGF parameters. We tested this on an independent, second version of the eye-movement task (version B; Fig. 1E; Supplementary Movie 3), in which the eye-movement trajectory was uncorrelated from the trajectory of the first version (version A) that was used to estimate the pEGF parameters.

For the reconstruction, we provided our algorithm with three sources of information: (1) the predicted responses of the pRF-only model for version B of the eye-movement task, (2) the pEGF parameters of each voxel, estimated from version A of the eye-movement task and (3) the obtained BOLD signal time series from version B of the eye-movement task. Importantly, the algorithm did not have access to the eye positions of version A or B of the eye-movement task. In brief, we divided the measured BOLD time series by the predicted time series from the pRF-only model to estimate the gain modulation over time. Then, we scaled each voxel's two-dimensional pEGF by the estimated gain and summed the scaled pEGFs of all voxels. We also computed a normalizer, which was the sum of all pEGFs scaled by the average gain at each time point. The scaled sum was then divided by the normalizer. After normalization, we estimated eye positions from the peak of the summed pEGFs (see Methods - Reconstructing eye position). For the reconstruction, we only included voxels where the pRF × pEGF model outperformed the pRF-only model for version A of the eye-movement task ($\Delta R^2_{adj} > 0$) and where the pRF-only model explained at least 10% of the variance in both versions of the eye-movement task ($R^2 = 0.1$).

The average reconstructed eye position is displayed in Fig. 4A. We assessed the quality of the reconstructed eye position by correlating it with the actual eye position using Pearson correlation (Fig. 4B). First, we cross-correlated the reconstructed and actual eye position with lags ranging from −10 to 10 TRs, as the reconstruction could lag the actual eye position because the gain is estimated from the unconvolved BOLD signal. The highest median correlation was obtained for a lag of 1 TR, equaling to 2 s (Supplementary Fig. 1G). This lag is similar to the lag we obtained using simulated time series, which indicates that the lag arises within our reconstruction algorithm (Supplementary Fig. 3). Thus, we used this lag in the rest of the analysis to assess the quality of the reconstructed eye position.

The median correlation across participants between reconstructed and actual eye positions for the horizontal component was $r_x = 0.77$ (inter-quartile range, IQR = 0.68–0.84), and for the vertical component $r_y = 0.64$ (IQR = 0.59–0.67) (Fig. 4C). To test the significance of these correlations, we repeated the reconstruction with permuted pEGF parameters, thus breaking the mapping between voxel's pRF and pEGF. In this permutation test, we included the same voxels as we did for the original reconstruction. For each reconstruction with permuted pEGF parameters, we computed correlations to obtain a null distribution of correlation values. Next, we computed $p$-values per participant and eye-position component (x, y) as the proportion of the null distribution that was larger than or equal to the true correlation (Fig. 4D). All $p$-values were <0.01 (see Supplementary Note 3), except for the vertical component of one participant ($p = 0.084$).

Because we—and others previously[20,21]—have been concerned that the presumed eye-position modulations may have resulted from unaccounted retinotopic stimulation, we simulated four scenarios, in which retinotopic stimulation was not completely accounted for, and/or the pEGFs were simulated to be non-existent. The simulation results indicate that the accurate reconstruction of eye position over time observed in our data most likely resulted from the presence of pEGFs, not from unaccounted retinotopic stimulation (for details and discussion, see Methods: Simulation 1: leftover retinotopic input and Supplementary Fig. 3).

As mentioned above, neurophysiological studies have demonstrated that the gain of visual responses is modulated by eye position in various visual areas. Our results also demonstrated that the pRF×pEGF model provides a better fit for the observed BOLD time series than the pRF-only model for all visual ROIs. To evaluate the quality of the pEGFs across the visual hierarchy, we assessed if and to what extent eye position can be reconstructed from single ROIs. We computed correlation estimates between reconstructed and actual eye positions by bootstrapping a fixed number of voxels ($\geq$150) per ROI. This ensures that potential differences in reconstruction quality are not due to the number of voxels used for the reconstruction. We combined the correlations for the horizontal and vertical eye-position components into a single, Fisher transformed average coefficient per ROI. Eye position could be reconstructed from all visual ROIs, but not from the visually non-responsive control DLPFC (Fig. 4E; Supplementary Note 4). Moreover, there was a significant difference in reconstruction quality between ROIs (F(9,90) = 38.03, $p < 0.0001$). The reconstruction quality of all ROIs was superior to that of the DLPFC. Interestingly, reconstruction quality was higher in V1, V2, V3, and V3AB than in the other ROIs (Fig. 4F). The reconstruction quality in V3AB is notably high, at the same level as early visual cortex and higher than the other visual ROIs such as V4, even though there was no significant difference in $R^2_{adj}$ between V3AB and V4 (Fig. 3B). A similar pattern was observed in a previous fMRI study that reconstructed static eye position from fMRI data[20]. Moreover, area V3A has been shown to compensate for pursuit eye movements in case of visual motion[22].

To summarize, we accurately reconstructed eye position over time by modeling contrast-based responses and implementing a multiplicative gain field. Successful reconstruction was possible throughout the human visual system, indicating EGFs are ubiquitous along the visual hierarchy in humans.

### Systematic relationship between pEGF centers and pRF eccentricity

Next, we explored the distributions and organization of pEGF parameters. Three distinct observations have been described in the literature regarding the organization of RFs and EGFs: (1) they have both been found to be contralaterally organized in V1[18] and V3A[12], (2) EGFs have been found to be ipsilateral, which in combination with contralateral RFs could give rise to a 'straight-ahead' bias in V1[17,33], and (3)

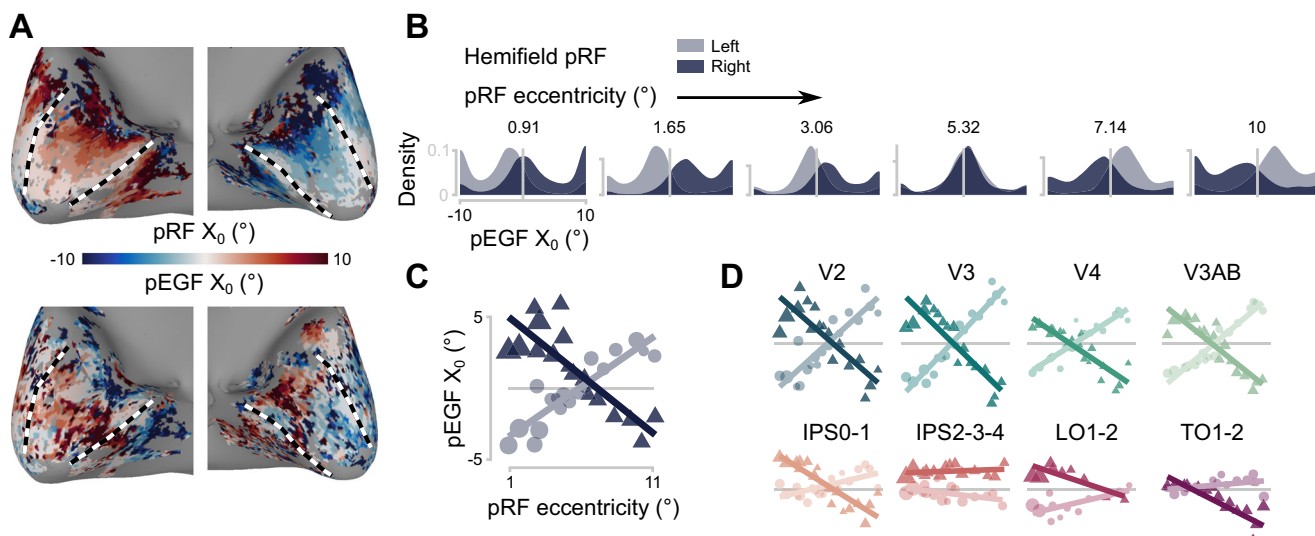

**Fig. 5 | Coupling between pRF center and pEGF center. A** Example surface maps of one participant. Black and white dashed lines mark the borders of V1. Top: horizontal component ($X_0$) of the pRF center, in degrees from the fovea. Bottom: $X_0$ of pEGF center, in degree from fixating straight ahead. See Figure S4 for surface maps of all participants. The pEGF $X_0$ shows a gradient from contralateral to ipsilateral along the posterior-anterior axis, the same direction as eccentricity of the pRFs. **B** Density distribution across all participants of pEGF $X_0$ binned for different pRF eccentricities in V1. Density estimates are split between pRFs in the left and right hemifield. At pRF eccentricities lower than -6°, pEGF centers are biased towards the side of the accompanying pRF. At higher pRF eccentricities this pattern reverses. **C** Median pEGF $X_0$ as a function of pRF eccentricity in V1. Colors follow the same conventions as in **B**. Circles and triangles represent the group median pEGF $X_0$ per pRF eccentricity bin and are scaled according to the number of voxels in that bin. Lines represent the least squares solution through the points. **D** Median pEGF $X_0$ as a function of pRF eccentricity for the other visual ROIs, all drawn to the same scale as in **C**. In all panels, the lighter color and circles represents pRFs in the left hemifield, the darker color and triangles represents pRFs in the right hemifield. See Figure S6A for the same plots for individual participants and S6C for the same results using a different configuration of the retinotopic representation of the visual input. Source data are provided as a Source Data file.

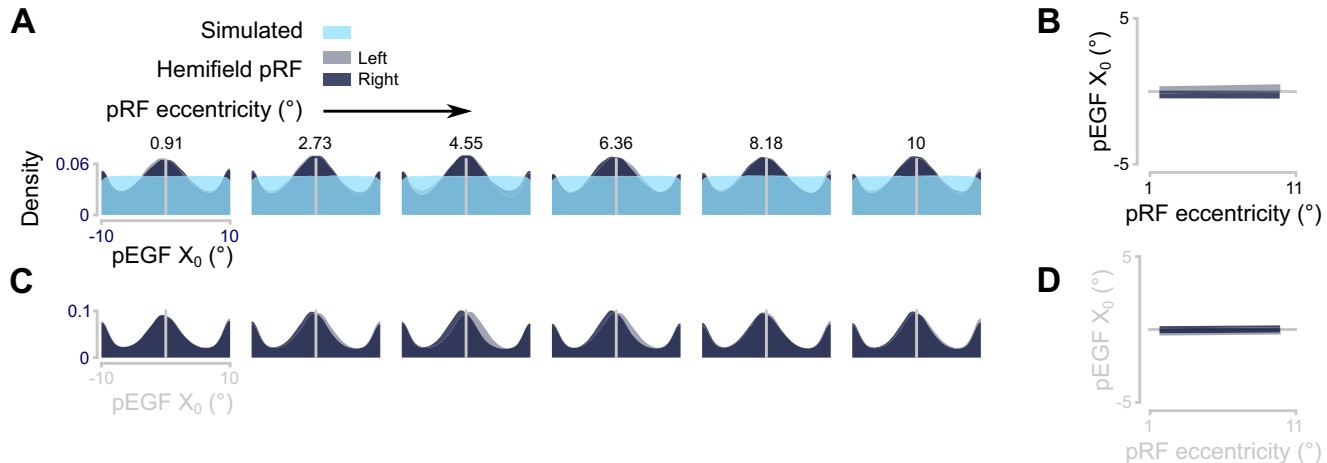

**Fig. 6 | Simulations of coupling between pRF center and pEGF center. A** Density distributions of simulated pEGFs ($X_0$), using the pRF parameters from each participant's V1. This simulation was performed to check for biases in our encoding model that would give rise to the features of the observed pEGF $X_0$ density distributions in Fig. 5B. In this simulation, pEGF parameters were randomly assigned to each pRF (light blue). Our encoding model estimated the pEGF parameters (dark blue). **B** Median estimated pEGF $X_0$ as a function of pRF eccentricity. Colors follow the same conventions as in **A**. Similar to Fig. 5C, we binned pRF eccentricity and computed the median pEGF $X_0$ per pRF eccentricity bin and are scaled according to the number of voxels in that bin. Lines represent the least squares solution through the points (data points are not plotted because they cause too much visual clutter around 0). In this simulation, no inversion like in Fig. 5C was present. **C** Like A, but in this simulation the pEGF was completely omitted, yet we asked our encoding model to estimate the pEGF parameters (dark blue). Two features are present in the distributions obtained with either simulation: a central peak and two peaks at the tails. Notably, the shift between pRFs in the left or right hemifield is not present. See Fig S5 for simulation results with different noise levels. $R^2_{adj}$ Like B but for the simulation where the pEGF was omitted (right). In this simulation, also, no inversion was present. Source data are provided as a Source Data file.

no organization is found at the population level in V6[11], LIP, area 7a[8], MT and MST[13]. In humans, evidence for a straight-ahead bias along the vertical axis was found in V1 and V2[21].

When we visualized the horizontal component ($X_0$) of the pEGF location we discovered a notable gradient in V1 from contra- to ipsilateral (Fig. 5A). This appears to follow a similar gradient as the eccentricity of pRFs. We quantified this gradient by plotting the density distributions of pEGF $X_0$ for different pRF eccentricities, where the density plots are split between pRF hemifield (Fig. 5B). There are three notable features in the distributions. First, there is a bias of pEGFs centers at central eye positions—note that this is different from the straight-ahead bias which refers to the location of pRFs at

straight-ahead, not gaze. Second, there seems to be a smaller bias at the two the tails of the distribution. Third, the center of density seems to be contralateral for pRFs with small eccentricities and shifts to ipsilateral for pRFs with an eccentricity of 6° to 8° from the fovea (see Supplementary Note 7).

The observed features in the distributions of pEGF parameters could arise from true underlying biological characteristics but could also be a consequence of the interactions between different parameters in our encoding model. Fox example, because our encoding model builds on pRFs, features of pEGF parameter distributions could be inherited from the distributions of pRF parameters, e.g., the relationship between pRF size and eccentricity[23]. To investigate such interactions in our encoding model, we examined which pEGF $X_0$ the model would yield when we simulated uniformly distributed pEGF $X_0$. In addition, we simulated which results we would obtain if there were no pEGFs driving our measured BOLD signals at all (for details, see Methods, Simulation 2: pEGF parameter distribution). From both simulations, we observed two of the three features: a bias at the center and a bias at the tails (Fig. 6A, C). The same pattern was observed for different noise levels (Supplementary Fig. 5). As such, it is likely that these two features arose from our modeling framework and do not represent biological features. However, the third feature, an inversion of pEGF laterality with pRF eccentricity, is present only in our data but not in the results from the simulation.

The inversion can be visualized more directly by binning voxels by pRF eccentricity and then showing the median pEGF $X_0$ per bin (Fig. 5C). There is a clear inversion in pEGF $X_0$ between pRFs in the left and right hemifield. This is not present in the results of the simulation (Fig. 6B, D). Visualizations for the other ROIs point towards a similar pattern (Fig. 5D). We tested the interaction between pRF eccentricity and visual hemifield per ROI using a linear mixed-effects model. For V1, V2, V3, V4, V3AB, IPS0-1, LO1-2, and TO1-2 there were significant interactions ($p < 0.0001$), but not for IPS2-3-4 and DLPFC ($p > 0.2$; see Supplementary Note 5).

Next, we examined the interaction between pRF eccentricity and the vertical component ($Y_0$) pEGF centers. The distribution of pEGF $Y_0$ was more heterogeneous across visual ROIs and participants (Supplementary Fig. 7). We ran a linear mixed-effects model of the interaction between pRF eccentricity and visual hemifield (upper/lower) per ROI. This analysis showed significant interactions for V1, V2, V3, and IPS0-1 ($p < 0.02$; see Supplementary Note 6), but the pattern of the interaction was different between these ROIs (Supplementary Fig. 7). As such, we are cautious in interpreting this heterogeneous interaction. It could be that the estimates of the vertical position are less reliable because the range of eye positions in the vertical direction (6.3°) was smaller than in the horizontal direction (12°).

To summarize, we observed a large-scale systematicity in the distribution of gain field centers at the population level that follows a topographical organization. We did not examine the size of the pEGF because we restricted the size to be larger than a voxel's pRF. As such, any systematicity in pEGF size might reflect a systematicity in pRF size, e.g., the known scaling of pRF size with eccentricity[23].

## Discussion

Here, we investigated the modulation of visual responses in human visual cortex as a function of eye position. We developed an encoding model that includes voxel-wise pRFs and population eye-position dependent gain fields (pEGF), probing the gain field mechanism in an active setting, while participants move their eyes and are presented with high contrast visual stimuli. First, we established that static stimuli elicit a visual response when they are brought into a pRF after a saccade, as expected from non-human primate data[27]. Then, we optimized the retinotopic representation of the stimulus to ensure that we accurately captured retinotopic visual responses. Inclusion of the multiplicative pEGF in the model improved the goodness of fit along

the human visual hierarchy, but not in a control area. As a more stringent validation, our encoding model allowed us to accurately reconstruct eye position over time. To test generalization and avoiding overfitting, model optimization and fitting were performed on one version of the task (task A) while reconstruction was performed on a different dataset, with independent eye trajectories (task B) and without further optimization.

The systematic characterization of pEGF properties was possible throughout the human visual system, also including the human parietal cortex. We examined the distribution of pEFG parameters at the population level and discovered a large-scale organization of pEGF centers: following the gradient of pRF eccentricity, pEGF centers shift from contralateral (for central pRFs) to ipsilateral (for peripheral pRFs).

An important aspect in studying gain modulations of retinotopic responses is to properly account for the retinotopic input. Failure to do so could result in true retinotopic activity being wrongly interpreted as the result of gain modulation. Previous fMRI studies that studied eye position responses or modulation of visual responses by eye position have been performed in complete darkness[34], with the use of motion-defined instead of luminance defined stimuli and covering the inside of the scanner bore with black felt[20] or only examining voxels responsive to a central region of the visual field[21,22]. Here, we took a different approach. We implemented a design where participants were actively performing eye movements in the scanner and factored peripheral visual stimulation into the retinotopic representation of our stimulus. We examined how the peripheral, static elements in the visual field should be incorporated into the representation. With this approach, we were able to adopt a relatively naturalistic eye-movement task for the participant, without the need to fixate at a single point for several minutes.

Next, it was necessary to exclude the possibility that our results were driven by biases introduced by our modeling strategy per se. For example, it could be that starting from uniform distributions of pEGF parameters, the modeling and fitting procedure would introduce distributional features not present at the input stage (Fig. 6A). Moreover, it is possible that our representation of the retinotopic input was either incomplete (i.e., some elements providing strong visual responses were omitted) or included elements that did not provide strong stimulation. To account for these possible confounds, we worked extensively on model simulations.

First, we assessed how eye-position reconstruction would be affected by leftover retinotopic activity (Supplementary Fig. 3). The result of this simulation showed that the most accurate reconstruction would be obtained if visual responses are truly modulated by eye position. Moreover, omitting several peripheral elements would barely affect reconstruction quality, but only if the visual responses are truly modulated by eye position. Indeed, in the simulations where we excluded pEGFs, reconstruction quality substantially decreased. Additionally, reconstruction from only left-over retinotopic input and without gain modulation was entirely unsuccessful (see Supplementary Fig. 3 for further discussion). Together, these simulations indicate that accurate reconstruction of eye position over time is tightly coupled to the existence of eye-position dependent gain modulation of visual responses and is not a result of unaccounted retinotopic stimulation.

This type of approach comes inevitably with a level of technicality, as we needed to formalize the model, the starting distributions, and the readout strategy. However, all these choices allowed for a careful disentanglement of genuine and spurious findings and were justified according to parsimony and first principles: the known underlying retinotopic organization of visual cortex and its response to contrast.

We observed a gradient in pEGF centers from contralateral to ipsilateral along pRF eccentricity, which was unlikely to have arisen from hidden interactions in our encoding model (as demonstrated by

the simulations). The gradient of pEGF centers is most prominent in early visual areas (V1, V2, V3), but is also present in higher areas (V3AB, V4). The gradient becomes less pronounced but is not absent further downstream the visual hierarchy (e.g., IPS0-1, LO1-2).

Although the modulation of visual responses by eye position has been demonstrated with neurophysiological recordings in many visual areas of non-human primates, there has been uncertainty whether EGFs are organized macroscopically (cf. retinotopic maps[3]), microscopically (cf. orientation columns[35]) or not at all[36]. In some recordings in V1[18] and V3A[12] a contralateral bias of EGFs has been observed, while others have observed an ipsilateral bias in V1[17,33]. This latter bias has also been measured with human fMRI[21]. For neurons with central RFs in V1, it is unclear why some have observed a contralateral bias[18], some an ipsilateral bias[33], and some no bias[17]. In most higher visual areas no prominent bias in EGF centers has been observed, including LIP and area 7a[8,36], MT and MST[13], V4[37], and V6[11]. It is important to note that the majority of these studies did not examine a relationship between RF location and EGF location, but rather examined the distribution of EGFs in their recorded sample of neurons. It is possible that an explicit relationship is only visible when EGF centers are expressed as a function of RF eccentricity. Otherwise, a bias in EGF centers would only be visible if the RFs of neurons in the recorded sample would be clustered around a single eccentricity. Moreover, both contra- and ipsilateral biases in EGF centers have been observed. Our data reconcile these findings with the demonstration of a topographic gradient of pEGF centers from contra- to ipsilateral following pRF eccentricity. Thus, corroborating the original idea of a topographic organization in gain-field position initially put forward in the discussion of Andersen and colleagues (1985).

Although there is currently no consensus as to why topographic maps exist in the brain, there are three non-exclusive hypotheses[38,39]. Firstly, it could be that throughout an animal's development, neighboring neurons are more likely to receive similar input as the result of chemical principles of axon guidance and local connections, thus giving rise to gradients in their later tuning profile. Secondly, topography might be metabolically and computationally efficient, limiting the need for long range connections to perform operations in a local sensory domain. Thirdly, the topography could be of functional relevance. In the case of EGFs, it has been suggested that ipsilateral EGFs for peripheral RFs lead to a "straight-ahead bias" which could serve prioritized processing of stimuli that are directly in front of an animal[40]. However, it is currently unclear whether this "straight-ahead bias" is related to the topographic relationship between pRF eccentricity and pEGF $X_0$ observed here.

Several candidate sources might underlie the modulatory signal that we observed along the visual hierarchy. A corollary discharge might inform the visual system about an impending eye movement, providing information about its amplitude and direction[41,42]. However, if the representation of eye position would depend exclusively on such a mechanism, the variability of the eye-position estimate would accumulate with each eye movement, making the system unstable over time[43]. Proprioceptive signals arising from the eye muscles reaching the visual cortex represent another source that could provide information about the current eye position[44]. Lastly, one possibility is that none of these signals are exclusively responsible for providing eye-position information to the visual cortex, but rather the signal is derived by integrating multiple sources[43]. Because in our task participants made an eye movement every few seconds and our measure is the BOLD signal, which evolves slowly over time, our data and the derived pEGF parameters could be influenced by both sources of eye-position information.

It has recently become clear that retinotopic responses in visual cortex are not only modulated by eye position[14-16,18] but also by the position of an animal in space[45,46]. Thus, gain modulation of topographically organized sensory neurons appears to provide a general principle by which sensory inputs from different sources are

transformed into a common reference frame. Uncovering the organizational principles of gain modulations can provide key insights into how this transformation can be so rapid and accurate, allowing animals to solve the ambiguity of sensory input during self-motion. Here, we provided evidence that EGFs are ubiquitous along the visual hierarchy in human neocortex and they exhibit remarkable organizational properties that follow a topographic organization.

## Methods

### Participants
Eleven healthy, human participants took part in this study after giving written informed consent (three authors and eight naïve, each scan session of approximately 2 h). All experimental procedures were approved by the local ethics committee at the School of Medical, Veterinary and Life Sciences of the University of Glasgow (reference number: 200180191 and GN19NE455). Participants completed at least two scanning sessions, with one additional training session being completed before the first scanning session, giving a total of 22 scanning sessions and 11 training sessions.

### Projector & eye tracker
Stimuli were projected at 120 Hz with a PROPixx projector (VPixx Technologies, Saint-Bruno, QC, Canada) onto a translucent screen (dimensions 320 × 400 mm) at the end of the scanner bore. Participants viewed the screen through an angled mirror at 8.89° visual angle from a distance of 959 mm (57.0 pixels/degree) and the infrared mirror of the eye tracker. Eye-position data were acquired with an MR compatible Eyelink 1000 at 250 Hz (SR Research, Ottawa, ON, Canada). Stimuli were presented with MATLAB (The Mathworks, Natick, MA) using the Psychtoolbox[47-49] and the Eyelink extension[50].

### pRF mapping – moving bar
Visual field mapping stimuli consisted of contrast-defined bars of cardinal and diagonal orientations, similar to those used in previous pRF mapping studies[23,51,52]. Participants fixated at a central gray fixation target ($r = 0.65°$; Thaler et al., 2013) while a high-contrast bar with a checkerboard pattern (-0.6° checks) swept across a uniform gray background in 20 equally spaced steps (Fig. 1E). Each step lasted 1.6 s. Bars had a width of 1.75°, exposing approximately three rows of checks. Alternating rows moved in opposite directions of each other with a speed of $0.5°s^{-1}$. Bars were presented with four different orientations and swept across the screen in eight different directions (two horizontal, two vertical, four diagonal). After 1 or 2 sweeps there was a baseline period of 12 s, in which no bars were presented, and participants kept fixation. The experiment included a total of 8 baseline periods. The total duration of the entire run was 310 s, during which 155 volumes were acquired. Participants completed at least 5 pRF mapping runs.

### pEGF mapping – active eye movements
Participants were instructed to follow a yellow fixation target across the screen with their gaze. The fixation target made 13 sweeps across the screen with 8 steps per sweep. Steps were between 1.28° and 1.40°, and lasted 1.5 or 1.67 s. Each sweep went from left to right or right to left, with some degree of variation over the vertical axis (Fig. 1E). There were 9 baseline periods of 20 to 25 s, one at the start of each run, one at the end, and seven in between. In addition to the fixation target, a bar with high-contrast, flickering, randomized checkerboard pattern was presented horizontally or vertically in the center of the screen, and/or diagonally in the corner of the screen. These bars were presented in various configurations at seven instances (Fig. 1E, Supplementary Movie 2 and 3). There were two versions of the task (version A and B). The eye-movement trajectories were uncorrelated between the two task versions (horizontal component: $r(177) = -0.07$, $p = 0.35$, vertical component: $r(177) = -0.09$, $p = 0.24$, Fig. 1E) and the bar configurations

were different. Participants completed at least 5 runs of version A and at least 4 runs of version B, in two scan sessions on different, non-consecutive days.

## Training session

Participants completed a behavioral training session before the scanning sessions to make them acquainted with the tasks. The training consisted of up to two runs of each task that would be performed in the scanner: (1) the moving bar, (2) active eye-movements version A and (3) active eye-movements version B. During the training, participants were seated in front of a BenQ XL2411 screen (540 × 300 mm) at a distance of 600 mm (39.6 pixels/degree). Their heads were stabilized using a chin-head rest. Eye movements were recorded with an Eyelink 1000 at 1000 Hz. Verbal feedback was provided to make sure participants were maintaining fixation when required and were not making anticipatory saccades in the eye-movement tasks.

## MRI – data acquisition

MRI data were acquired with a 7 T Siemens Magnetom Terra system (Siemens Healthcare, Erlangen, Germany) and a 32-channel head coil (Nova Medical Inc., Wilmington, MA, USA) at the Imaging Centre of Excellence (University of Glasgow, UK). We collected T1-weighted MP2RAGE anatomical scans (anatomy) for each participant (0.625 mm isotropic, FOV = 160 × 225 × 240 mm$^3$, 256 sagittal slices, TR = 4.68 ms, TE = 0.00209 ms, TI = 0.84 ms, TI1 = TI2 = 2.37, flip angle 1 = 5, flip angle 2 = 6, bandwidth = 250 Hz/px, acceleration factor = 3 in primary phase encoding direction).

During the tasks, functional data were acquired as T2*-weighted echo-planar images (EPI), using the CMRR MB (multi-band) sequence with the following acquisition parameters: resolution = 1.5 mm isotropic, FOV = 192 × 192 × 84 mm$^3$, repetition time (TR) = 2000 ms, echo time (TE) = 25 ms, flip angle = 72°, multiband acceleration factor = 2, phase encoding direction = anterior to posterior. For the pRF mapping task we collected 155 volumes per run (310 s), and for the eye-movement tasks (versions A and B) 179 volumes (358 s). For each MRI session, we recorded 5 volumes with the same EPI sequence parameters with the phase-encoding direction inverted (posterior to anterior; top-up EPI) to correct for susceptibility-induced distortions and facilitate co-registration with the anatomical data. We acquired the top-up EPIs at the beginning of the MRI session (between the first and the second EPI run) and at the end of the MRI session (before the final EPI run).

## MRI – EPI preprocessing and coregistration

All analyses were performed in AFNI, R and Python (Nighres)[53–56]. Functional scans were slice timing corrected using the AFNI function 3dTshift.

For each MRI session, we computed a warp field to correct for geometric distortions from the original (non-motion corrected) EPI volumes, using the function 3dQwarp: we averaged the first 5 volumes of the EPI following the acquisition of the top-up EPI, and averaged the 5 volumes of each top-up EPI run. The resulting undistorted (warped) EPI volume is the halfway warping between the two average volumes[57].

Motion parameters between runs in a session were estimated by aligning the EPI volumes to the first volume of the first EPI run using the function 3dvolreg. To minimize interpolation applied to the EPI data, the motion estimates and warp field results were combined and applied in a single step, using the function 3dNwarpApply. Then, we computed the motion correction and warped mean EPI volume of the MRI session by averaging over all warped and motion corrected EPI volumes between runs and collapsed over all time points in the 'moving bar' paradigm. This resultant mean EPI volume was co-registered to the anatomy. First, we brought the anatomy and the session mean EPI volume into the same space by aligning their respective centers of mass. Next, the 'Nudge dataset' plugin in AFNI was used to manually provide a good starting point for the automated

coregistration. This registration consisted of an affine transformation, using the local Pearson correlation as cost function[58] in the function 3dAllineate. The individual motion-corrected runs were then de-spiked (using the function 3dDespike), scaled to obtain percentage BOLD signal change, and detrended with a 3$^{rd}$ order polynomial (using the function 3dDetrend). We averaged all processed EPI runs per task and participant to increase the signal to noise ratio.

EPI volumes from the active eye-movement task A and B were co-registered to the EPI volumes of the moving bar to obtain a voxel-by-voxel correspondence between tasks. The outcome of the coregistration for each participant and session was visually checked by evaluating the location of anatomical markers as gray matter/white matter (GM and WM, respectively) and GM/cerebro-spinal fluid (CSF) boundaries in the calcarine sulcus and the parietal cortex. All analyses on the functional data were performed on GM voxels in EPI space. For each voxel in the GM mask, we computed its normalized cortical depth, ranging from 0 (GM/WM surface to 1 (GM/cerebro-spinal fluid surface), based on a volumetric model[59]. We excluded the inner and outer 10% of GM to exclude effects of partial volume. EPI time series were smoothed at the single voxel level with a 3-point Hamming window over time. No spatial smoothing was applied.

## MRI – anatomy segmentation

The GM and WM were automatically segmented based on the anatomical scans using in-house software. Segmentation was performed on a downsampled resolution of the anatomical data (0.7 mm, isotropic, downsampling performed with the function 3dResample). T1-w images were co-registered to an atlas[60] to remove the cerebellum and sub-cortical structures. We separated the T1-w images in 6 different sections from posterior to anterior, with each section being used separately as an input for the 3dSeg function in AFNI to isolate the WM. The WM masks obtained from each part were summed together resulting in a whole brain WM mask. To derive the GM segmentation, we started from the obtained whole brain WM segmentation and an atlas co-registered to the T1-w images to acquire 35 regions per hemisphere[60]. Next, a distance map from the WM/GM boundary to the pial surface was built computing the Euclidean distance of each voxel from the WM/GM border. Negative distances were assigned inside the WM and positive distances were assigned from the WM borders outwards. For each region, the coordinates were divided into four separate subparts using k-means clustering. This step was necessary to accurately delineate the boundaries on small portions of each region (subregion), with a highly homogeneous T1-w signal. For each subregion, voxels within −2 and 7 mm from the WM/GM border were selected and their T1-w intensity was stored for further analysis. For each region's subregion, we obtained 10 bins between −2 and 7 mm from the WM/GM border. For each bin, we computed the inter-quartile estimate of T1-w intensity. We calculated the 75% quantile of the inter-quartile estimates and computed the associated Euclidian distance from the WM/GM border. This Euclidean distance was taken as the cortical depth associated with the subregion. To improve the obtained GM segmentation, the WM and GM masks were fed to the "Cortical Reconstruction using Implicit Surface Evolution" (CRUISE) algorithm in Nighres.

## Population receptive fields

PRFs of voxels in the GM mask were estimated as 2D isotropic gaussians with compressive spatial summation[23,24].

$$pRF(x_r, y_r) = e^{-\frac{(x_r - x_{r0})^2 + (y_r - y_{r0})^2}{2\sigma_r^2}} \quad\quad (1)$$

$$size = \frac{\sigma_r}{\sqrt{n}} \quad\quad (2)$$

The pRF (Eq. (1)) is defined as a function of retinotopic coordinates $(x_r, y_r)$ with four parameters: horizontal center $(x_{r0})$, vertical center $(y_{r0})$, standard deviation $(\sigma_r)$, and compressive component (n). The compressive component, n, is an exponent ($\leq 1$) that is applied after multiplying the pRF with the stimulus $(S_{bar})$ and summing over x and y (Eq. (3)). The size of the pRF is defined with both $\sigma$ and n (Eq. (2)).

$$r(t) = \left[ \sum_{x_r, y_r} pRF(x_r, y_r) S_{bar}(x_r, y_r, t) \right]^n \qquad (3)$$

To create a predicted response (r), the pRF is multiplied elementwise with a spatially (12 pixels/°) and temporally (6 Hz) downsampled version of the moving bar stimulus ($S_{bar}$; Supplementary Movie 1), binarized to a contrast-defined image (Eq. (3)).

$$\hat{y}(t) = r(t) * h(t) \qquad (4)$$

We convolved the predicted response (r) for the moving bar stimulus with a standard hemodynamic response function (HRF; $h_b$; $n_1 = 6.0$, $t_1 = 0.9$ s., $n_2 = 12.0$, $t_2 = 0.9$ s., $a_2 = 0.35$;[61]) to get a predicted BOLD response ($\hat{y}$, Eq. (4)). After the convolution, the predicted BOLD time series ($\hat{y}_{aA}$) were further downsampled to match the TR (0.5 Hz). We estimated the pRF parameters for each voxel $(x_0, y_0, \sigma, n)$ by taking the prediction that yields the maximum variance explained (R2). Parameters were found using an exhaustive grid search over 5400 predefined parameter combinations (12 $x_{r0}$, 9 $y_{r0}$, 10 $\sigma_r$, 5 n).

### ROI definition

After the pRF analysis, we plotted the resulting polar angle map on each participant's inflated hemisphere and identified the regions of interest (ROIs) based on the reversals in polar angle of visual field position preference[3,62,63]. In the current study, we selected the following ROIs: V1, V2 (combination of V2v and V2d), V3 (combination of V3v and V3d), V3A and V3B combined (V3AB), V4, lateral occipital cortex 1 and 2 combined (LO1-2), temporal occipital cortex 1 and 2 combined (TO1-2), intraparietal sulcus areas 0 and 1 (IPS0-1, combination of IPS0 and IPS1) and higher intraparietal sulcus areas insofar as they were defined by the angle maps (IPS2-3-4, this included IPS5 for some participants). In addition to these visual areas, we also defined an area that had no clear visual response (i.e., low variance explained by the best fitting pRF), located in the dorsolateral prefrontal cortex (DLPFC) and that approximately matched the size of the primary visual cortex (V1).

ROIs were initially defined on the surface and projected into the underlying anatomy using the function 3dSurf2Vol. ROIs were drawn by hand on a high-resolution anatomical surface which was computationally inflated using SUMA[64]. Anatomical ROIs were projected into the EPI volume of the moving bar task by inverting the co-registration affine transformation (see section: EPI preprocessing and MRI – EPI co-registration)

### Retinotopic representation of visual input in eye-movement tasks

Before modeling pEGFs, we captured the visual input in the active eye-movement task (A and B) that was elicited by the retinotopic stimulation of each voxel's pRF. We created retinotopic, contrast-defined representation of the visual input to model the data from the active eye-movement tasks. These representations were downsampled to 6 Hz, and a resolution of 6.7 pixels/deg. Because the eyes of the participants were following the fixation target, we centered the stimulus around the fixation target with a lag of one frame (166 ms) to capture the median saccade latency (135 ms, 141 ms; Supplementary Fig. 1D). We included various elements in the participant's peripheral visual field based on a sketch of the view from inside the scanner (Fig. 1F; Supplementary Movie 2). These objects included the screen edge, the

coil-mirror, the eye tracker, and the bore. The dimensions of the field of view of the modeled stimuli were 40° × 22°.

To evaluate to what extent the estimated pRFs could be translated to the active eye-movement task, we computed the variance explained of a pRF-only model to various configurations of the contrast-defined stimulus (Fig. 2). This also allowed us to examine how the contrast-defined retinotopic visual input should be represented before we continued to include the pEGFs into the models.

We considered two ways by which the elements within the scanner bore (and the bore itself) could provide visual stimulation in addition to the checkerboard bars. First, they could provide a transient input after saccade offset that is similar to a stimulus onset during fixation[27]. Second, the elements could provide input continuously, as a result of drift and microsaccades[65]. Although both effects have been observed in neurophysiological recordings, we do not know whether they are reflected in BOLD responses, and we do not know to what degree they elicit visual responses relative to the response generated by the flickering checkerboard bars (which provide a strong visual stimulus). For example, the elements could generate a strong transient response after saccade offset and only a weak response during fixation, but in both cases, they only generate a response smaller than the response elicited by the flickering bars. To find the optimal representation, we examined the variance explained of different versions of the stimulus model where we systematically varied (1) the contrast of the elements after a saccade and during fixation, relative to the contrast of the bars, and (2) the duration of the postsaccadic window in which the elements should be considered.

We first examined the contrast levels of the peripheral elements after a saccade and during fixation. The contrast of the elements in the retinotopic representation was set to a value between 0 (absent) to 100% (equal to bars) for 2 frames (333 ms) after a saccade (with the levels 0, 5, 10, 20, 50, 90, 100%). In the remaining frames, during fixation, the contrast was independently varied between 0 and 100% (same levels). For each of the 49 contrast combinations (7 contrast levels after a saccade, 7 contrast levels during fixation), we generated predicted BOLD time series based on: (1) that particular stimulus model, (2) the trajectory followed by the eyes in the active eye-movement task A, and (3) each voxel's pRF estimated from the moving bar experiment. We then fitted the predictions to the data using ordinary least-squares regression. For all further analyses, we used the contrast configuration that yielded the highest R2 in V1 voxels. The best configuration was at a fixation contrast = 0% and a postsaccadic contrast = 100% (Fig. 2D).

Next, we examined the duration of the postsaccadic contrast increase. We varied the duration of the postsaccadic contrast increase of the peripheral elements from 166 ms (1 frame at 6 Hz) to 5333 ms (32 frames), with the levels 1, 2, 3, 4, 6, 8, 12, 16, 24 and 32 frames. Following the same approach as described above, we generated predictions from the combination of each voxel's pRF and the different configurations of the retinotopic representation. For all further analyses, we used the contrast configuration of 500 ms (3 frames), which approximated well the model fits in early visual cortex (Supplementary Fig. 1F).

Following this strategy, we accounted for the signal variability in the active eye-movement task (A and B), elicited by the retinotopic stimulation of each voxel pRF while participants were performing eye movements within the scanner.

### Population eye-position dependent gain fields

We modelled eye-position dependent gain fields (pEGF) as 2D isotropic gaussians:

$$pEGF(x_e, y_e) = a e^{-\frac{(x_e - x_{e0})^2 + (y_e - y_{e0})^2}{2\sigma_e^2}} + (1 - a) \qquad (5)$$

Each pEGF was described as a function of eye-position coordinates $(x_e, y_e)$ with four parameters: horizontal eye position $(x_{e0})$, vertical eye position $(y_{e0})$, standard deviation $(\sigma_e)$, and amplitude $(a_e)$, with respect to the center of the screen (Eq. (5)).

$$g(t) = pEGF(x_{eyeA}(t), y_{eyeA}(t)) \tag{6}$$

Gain modulation (g) was obtained by passing the eye positions during version A of the active eye-movement task $(x_{eyeA}, y_{eyeA})$ through the pEGF (Eq. (6)). Note that the pEGF of each voxel was estimated on top of the already estimated pRF.

$$r(t) = \left[ \sum_{x_r, y_r} pRF(x_r, y_r) S_{eyeA}(x_r, y_r, t) \right]^n \times g(t) \tag{7}$$

The predicted response in the active eye-movement task A (Eq. (7)) is the product of the predicted response from the pRF to the down-sampled, binarized version of the active eye-movement stimulus $(S_{eyeA}$, see section Retinotopic representation of visual input during active eye-movement task) and the gain modulation generated by the pEGF (g).

$$\hat{y}(t) = r(t) * h(t) \tag{8}$$

The predicted response (r) was convolved with the standard HRF (h, Eq. (8)). After the convolution, the predicted BOLD time series (ŷ) were further downsampled to match the TR (0.5 Hz). We estimated the pEGF parameters for each voxel $(x_{e0}, y_{e0}, \sigma_e, a_e)$ by taking the prediction that yielded the maximum variance explained (R2). Parameters were found using an exhaustive grid search over 3888 predefined parameter combinations (12 $x_{e0}$, 9 $y_{e0}$, 6 $\sigma_e$, 6 $a_e$).

### Reconstructing eye position

If the pEGF parameters that we had estimated truly correlate with eye position and are not resulting from a confound in the time series of our task, then the pEGF parameters should be informative about eye position in a different context (i.e., generalize to a different eye-movement task). After we fitted pEGF parameters to the data of version A of the active eye-movement task, we reconstructed eye position from version B. Please note that eye-position trajectories were uncorrelated between the two versions of the eye-movement task and the bar configurations were different (see section 'pEGF mapping–active eye-movements'; Fig. 1E).

As a starting point for the reconstruction, we assumed the measured BOLD response in each voxel would be generated by the response of the underlying neural population (r). The population response can in turn be characterized by the product of its pRF with the stimulus $(S_{eyeB})$, multiplied with an eye-position dependent gain factor (g, Eq. (9)):

$$r(t) = \left[ \sum_{x_r, y_r} pRF(x_r, y_r) S_{eyeB}(x_r, y_r, t) \right]^n \times g(t) \tag{9}$$

From this definition, we can estimate the gain factor (ĝ) for every time point (Eq. (10)).

$$\hat{g}(t) = \frac{r(t)}{\left[ \sum_{x_r, y_r} pRF(x_r, y_r) S_{eyeB}(x_r, y_r, t) \right]^n} \tag{10}$$

However, because our measured signal is the BOLD signal, not the direct population response (r), we approximated the gain factor with an HRF-convolved version of the denominator (Eq. (11)).

$$\hat{g}(t) \approx \frac{y(t)}{\left[ \sum_{x_r, y_r} pRF(x_r, y_r) S_{eyeB}(x_r, y_r, t) \right]^n * h(t)} \tag{11}$$

y(t) is the measured BOLD response in version B of the active eye-movement task and h(t) is the standard HRF. We accounted for the retinotopically elicited response by first bounding both the data (y) and the pRF-only prediction between 0 and 1. Then we linearly scaled the predicted time series (y) based on the pRF-only model. Finally, we divided the data by the scaled predicted time series from the pRF-only model to estimate ĝot). We removed values falling below the 1st or above the 99th percentile of each voxel's ĝ and set them to range within the 1st or 99th percentile of the values (with an absolute minimum and maximum of 0.05 and 2.5).

Next, we scaled the 2D pEGF with the estimated gain (ĝ) for every time point (Eq. (12)). We then summed the scaled pEGFs of all voxels (N) and divided that sum by the sum of all pEGFs scaled by the average gain across voxels (ḡ) at that time point (Eqs. (13) and (14)). For each time point, we estimated the eye position (x̂, ŷ) to be the location where the scaled sum of all pEGFs was maximal (Eqs. (15) and (16)).

$$\text{scaled}(x_e, y_e, t) = \sum_{i=1}^{N} \hat{g}_i(t) pEGF_i(x_e, y_e) \tag{12}$$

$$\text{unscaled}(x_e, y_e, t) = \bar{g}(t) \sum_{i=1}^{N} pEGF_i(x_e, y_e) \tag{13}$$

$$f(x_e, y_e, t) = \frac{\text{scaled}(x_e, y_e, t)}{\text{unscaled}(x_e, y_e, t)} \tag{14}$$

$$\hat{x}(t) = \max_{j \in x_e} f(j, y_e, t) \tag{15}$$

$$\hat{y}(t) = \max_{k \in y_e} f(x_e, k, t) \tag{16}$$

Note that with this approach eye position is inferred from the estimated gain and pEGF. Eye position cannot be estimated from a single pEGF because its maximal value would always be the center of the 2D gaussian that defines the pEGF. The potential precision of the reconstructed eye position depends on the number of pEGFs used, provided that their parameters are uniformly distributed over the parameter space. This principle resembles eye-position reconstruction from single unit recordings[32,36]. For the reconstruction of eye-movement trajectory (of version B of the eye-movement task) we only included voxels where:

1. $R^2_{adj}$ (Eq. 17) for the pRF × pEGF model > $R^2_{adj}$ for the pRF-only model for version A of the active eye-movement task. Where $R^2$ is the variance explained, n is the number of time points in the BOLD time series, k is the number of parameters. This measure of variance explained accounts for the number of parameters in the different models.

$$R^2_{adj} = 1 - \left[ \frac{(1 - R^2)(n-1)}{n - k - 1} \right] \tag{17}$$

2. the pRF-only model yielded $R^2 > 0.1$ for version A of the active eye-movement task.

3. the pRF-only model yielded $R^2 > 0.1$ for version B of the active eye-movement task.

In addition to examining whether eye position can be reconstructed at all, i.e., using all voxels in the visual cortex, we repeated the reconstruction for individual ROIs (see Statistics). We applied the same voxel inclusion criteria for the visual ROIs. For the control area in DLPFC, we only used non-visually responsive voxels, i.e., where both the pRF-only and the pRF×pEGF model yielded $R^2 < 0.1$. The reason for using different selection criteria for the control area is to include a non-visual area and check whether any artifacts that are unknown to us could have driven the results we obtained from the visual areas. If we were to apply the same selection criteria for the DLPFC, we would include on average 1 voxel per participant (0.03%). Such a low number of voxels would compromise any possibility to reconstruct eye position over time. For the other visual areas, the number of included voxels ranged between 344 (V4) and 1172 (V1) voxels, or in percentages, between 24.5% (IPS2-3-4) and 43.1% (V4). With the separate inclusion criteria for DLPFC, we included on average 924 voxels (34.8%), which is comparable to the number of voxels included in the visual areas, thus making the comparison fair.

## Statistics

1. Linear mixed-effects model of change in $R^2_{adj}$ per ROI
   We analyzed the median difference in $R^2_{adj}$ between the pRF-only and pRF×pEGF models using a linear mixed-effects model (Eq. 18). The analysis we performed in the same manner as described above. For this analysis, we selected all voxels in the upper half of a median split based on the $R^2$ from the moving bar paradigm, i.e., how well the pRF described the time series of the moving bar paradigm. Hence, we use those voxels whose time series are relatively well characterized by the pRF. From those voxels, we computed the median $R^2_{adj}$ per ROI (i) and participant (j). The medians were analyzed with the mixed-effects model.

$$\Delta R^2_{adj\,ij} \sim \beta_0 + \beta_i \mathbf{X} + \mathbf{Z}_j \qquad (18)$$

   $\beta_0$ represents the $R^2_{adj}$ in V1 (set as the reference ROI), $\beta_i$ are the differences in $R^2_{adj}$ (with one coefficient for each row in the design matrix), $\mathbf{X}$ is the design matrix with a column for each ROI, and $\mathbf{Z}$ is the subject-specific deviation from the intercept ($\beta_0$). This model was compared to a 'null model' where the term $\beta_i \mathbf{X}$ was removed. Models were fitted using the "lme4" package in R[66]. The significance of main effects was tested using the 'anova' function. We compared differences in $R^2_{adj}$ between ROIs and against a correlation of 0 using parametric bootstrapping ($N = 10^6$) with the "bootMer" function of the "lme4" package. Statistical inferences were based on the 95% confidence intervals, corrected for multiple comparisons with the Bonferroni correction.

2. Reconstruction correlation coefficients and *p*-values per participant
   We used Pearson correlation between the position of the fixation target and the reconstructed eye positions to determine whether reconstruction was successful. Correlations were computed for the horizontal (x) and vertical component (y) separately. In addition, we computed a cross-correlation with different lags to account for potential delays in the reconstructed eye position introduced by the HRF.
   To test whether the fitted parameters were essential for eye-position reconstruction, we repeated the reconstruction 1000 times, but with permuted pEGF parameters. The pEGF parameters of all voxels that met the inclusion criteria were shuffled before reconstructing the eye position and computing the correlations with the actual position of the fixation target. We computed p values as the proportion of reconstructions that yielded a higher correlation between the reconstruction and the

actual eye position than the same correlations obtained with the non-permuted pEGF parameters.

3. Linear mixed-effects model of average correlation per ROI
   With our reconstruction method, reconstruction accuracy increases with an increased number of voxels (see Reconstructing eye position). To fairly compare the quality of eye-position reconstruction, we needed to equate the number of voxels used for the analysis across ROIs. For each ROI, we computed a distribution of correlation coefficients between the reconstructed eye position and the position of the fixation target by bootstrapping a fixed number of voxels from each ROI (90% of the smallest ROI per participant, with a minimum of 150 voxels; N bootstrap = 1000). We took the Fisher-transformed correlation coefficient (averaged between horizontal and vertical components) per ROI and participant and compared these with a linear mixed-effects analysis on the group level (Eq. 19).

$$\frac{\left[ \operatorname{arctanh}(r_x) + \operatorname{arctanh}(r_y) \right]_{ij}}{2} \sim \beta_0 + \beta_i \mathbf{X} + \mathbf{Z}_j \qquad (19)$$

   where, $r_x$ and $r_y$ are the correlation coefficients of the horizontal and vertical components of the eye position and fixation target positions, per ROI (i) and participant (j). Further inferences were drawn similarly to the method described in part 1 of this section (Linear mixed-effects model of change in R2adj per ROI).

4. Linear mixed-effects model of pEGF $X_0$ as a function of eccentricity and hemifield.

For each participant, we selected the best half of the voxels per ROI, where 'best' is defined as the highest variance explained by the pRF × pEGF model. From those voxels, we binned the pRF eccentricity and computed the average pEGF $X_0$. This was performed for each visual hemifield separately. Next, we constructed a linear mixed-effects model for each ROI separately (Eq. (20)). We did not include ROI as a fixed effect to keep the coefficients interpretable.

$$\text{pEGF } X_j \sim \beta_0 + \beta \mathbf{X} + \mathbf{Z}_j \qquad (20)$$

where, $\beta_0$ represents the pEGF $X_0$ in the left hemifield at an eccentricity of, $\beta$ are the three fixed-effect coefficients for the effects of hemifield, pRF eccentricity, and the interaction between the two. $\mathbf{X}$ is the design matrix with a column for the fixed effect, and $\mathbf{Z}$ is the subject-specific deviation from the intercept ($\beta_0$). For each ROI model, we compute the significance of the interaction (and other two fixed effects using the 'anova' function.

## Simulation 1: leftover retinotopic input
With this simulation, we examined to what extent leftover retinotopic stimulation could explain the reconstruction results that we obtained. The simulation consisted of four steps:
1. simulate BOLD time series
2. add noise
3. recover pEGF parameters from simulated time series
4. reconstruct eye position using the recovered pEGF parameters

The pRF and pEGF parameters (if applicable) were obtained from the estimates in the real data. This way any implicit relationships between pRF parameters or between pEGF and pRF parameters remained intact. We only used parameters from voxels which met the same three inclusion criteria as for the actual reconstruction:
1. $R^2_{adj}$ (Eq. (17)) for the pRF × pEGF model > $R^2_{adj}$ for the pRF-only model for version A of the active eye-movement task
2. the pRF-only model yielded $R^2 > 0.1$ for version A of the active eye-movement task.

**Table 1 | Simulation scenario's**

| pEGFs exist | Peripheral stimuli generate visual response in simulation | Peripheral stimuli generate visual response in retinotopic representation used to model pEGF parameters |
|---|---|---|
| Yes | Yes | Yes |
| Yes | Yes | No |
| No | Yes | No |
| No | No | Yes |

3. the pRF-only model yielded $R^2 > 0.1$ for version B of the active eye-movement task.

We simulated four time series for both versions of the eye-movement task using different configurations of pEGF existence, peripheral stimuli strength, and correct or incorrect model assumptions about the peripheral stimuli (see Table 1).

These scenarios can be summarized as:

1. pEGFs exist and retinotopic representation is accurate
2. pEGFs exist but there is leftover retinotopic stimulation
3. pEGFs do not exist and there is leftover retinotopic stimulation
4. pEGFs do not exist and retinotopic representation includes non-existing elements

For each participant, these four simulation scenarios were repeated 100 times with different renditions of noise. The noise was generated using the "neuRosim" package in R. We used a combination of physiological noise, system noise and task noise (which depended on retinotopic stimulation in the pRF of a voxel). The ratio of the noise components was set to 1:1:0.1, respectively. The sum of these components was scaled to have a total amplitude of 0.1 (all simulated time series were scaled between 0 and 1). From these noisy time series we estimated pEGF parameters, irrespective of whether they were simulated to exist or not. These estimations were performed using the same model as we used for the actual data. Finally, the reconstruction was also performed using the same method as the actual data. Detailed results of this simulation are reported in the caption of Figure S2.

## Simulation 2: pEGF parameter distribution

With this simulation, we examined whether our modeling framework would yield biased pEGF parameters as the result of any unforeseen interactions. For this simulation, we used parameters from the pRF fits, such that implicit relationships between pRF parameters would also be included in the simulation. We selected voxels from V1 where the pRF explained at least 25% of the variance in the moving bar paradigm (i.e., that data to which the pRFs were fitted; pRF-only model).

The simulation consisted of seven steps:

1. create contrast-driven time series based on the pRFs based on the retinotopic representation of visual input of version A of the eye-movement task.
2. randomly assign pEGF parameters to each voxel, with the restriction that the s.d. of the pEGF is larger than the s.d. of the pRF
3. modulate the contrast-driven time series with the assigned pEGF
4. convolve the time series with the hemodynamic response function
5. scale time series between 0 and 1
6. add noise using from the 'neuRosim' package. We used a combination of physiological noise, system noise, and task noise (which depended on retinotopic stimulation in the pRF of a voxel). The ratio of the noise components was set to 1:1:0.1, respectively. The noise was scaled the s.d. of the noise to 0.05, 0.1, and 0.2.

7. estimate pEGF parameters using the same framework as we used for the real data.

The parameters that were estimated from the simulated data were analyzed and visualized in the same manner as described for the actual data.

### Reporting summary

Further information on research design is available in the Nature Portfolio Reporting Summary linked to this article.

## Data availability

All data are stored by the authors and will be made available upon reasonable request. Source data are provided with this paper and are available on Open Science Framework (https://doi.org/10.17605/OSF.IO/GTD5R). Source data are provided with this paper.

## Code availability

All scripts used to run control the visual presentation of the stimuli and the scripts to perform the analyses are available on Open Science Framework (https://doi.org/10.17605/OSF.IO/GTD5R).

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

## Acknowledgements

We are grateful to A. Tyler Morgan for their guidance and expertise in the development of the sequences, making and breaking the eye-tracker equipment into the scanner, and their assistance during the scanning sessions. We also thank Nils Nothnagel for his vital assistance during the scanning sessions, Frances Crabbe for her endless support with the collection of anatomical scans and Manuela Ruzzoli for her keen comments on the analysis. K.M. is supported by the Doctoral Training Programme (DTP) from the College of Medical, Veterinary and Life Sciences (MVLS), University of Glasgow. A.F. is supported by a grant from the Biotechnology and Biology research council (BBSRC, grant number: BB/S006605/1) and the Bial Foundation, Bial Foundation Grants Programme Grant ID: A-29315, number: 203/2020, grant edition: G-15516.

## Author contributions

J.H.F. and A.F. designed the experiments. J.H.F., K.M. and A.F. collected the data. J.H.F. performed the analyses. A.F. guided the analysis. J.H.F., K.M. and A.F. wrote the manuscript.

## Competing interests

The authors declare no competing interests.
