## [Peer Review File · Nature Communications]

Topographic organization of eye-position dependent gain fields in human visual cortexREVIEWER COMMENTS

Reviewer #1 (Remarks to the Author):

The authors conducted three human fMRI experiments with and without saccadic eye movements over moving or static-flickering bar-graph stimuli in order to systematically model the influence of eye positions on the fMRI responses to visual bar stimuli. Using extensive modelling of population receptive fields (pRFs) as well as population eye position gain fields (pEGFs) the authors show that eye-positions indeed modulate neural responses in a systematic way, and that pEGFs centers tend to anti-correlate with eccentricity of the pRFs.

Overall, this is a well conducted, well presented, highly readable, timely, and important paper that will receive considerable attention from both, visual, motor, and theoretical neuroscientists. The methods are sound and well documented.

While I recommend publication, I have several major and minor points regarding some aspects of modelling, presentation of statistics, and concepts in the introduction.

Major:

1) Introduction:

I like that you try to introduce the reader from scratch to the topic. However, it is important to not conflate concepts and keep things simple. Your art example is beautiful, but why then refer to another piece of art that is not shown (Pollock)?

L32ff: "With only the retinotopic input at our disposal, we would quickly lose track of all the stars and other elements. As a result, "La nuit étoilée" would become barely indistinguishable from Jackson Pollock's "Night mist"."

This statement lacks conceptual motivation, for two reasons:

First, you introduced the eye gain field problem in terms of attribution of movement (self vs other), not in terms of improving recognition of images.

Second, also a brief fixation (with no eye movements) will allow readers to see and differentiate between two paintings.

Suggestion: cut L32ff.

Results

2) L134 ff

The approach of including peripheral visual cues outside the screen is interesting, but requires more detailed description and statistics in order demonstrate its usefulness (or lack thereof) (both would add important knowledge to the field).

2a) You go to great lengths to model and maximize the variance explained by not only modelling the on-screen stimulus, but also the static peripheral cues (screen border etc). Fig 2 quantifies several parameters, the main text even more. However, in the end, the reader is left uncertain about the benefits of this is approach: the model fits best when “during fixation” the peripheral contrast is 0. The effects “after saccade” are overall small. This leaves me wonder: does inclusion of the periphery make a statistically significant difference, and how much, compared to leaving it out completely? How large is this effect across cortical regions? Please provide a plot of optimal modelling peripheral contrast vs not at all, comparing the respective R square, with statistical tests (F-tests?), across visual regions (analogue to Fig3B). This would be useful information to inform future experiments.

2b) L134ff: “The median variance explained”: it is unclear what this refers to: the best-fitting model, or the additional variance explained by including the peripheral static field? Please clarify.

3) Fig 3B: if I read the sup mat “statistics output” correctly, then all ROIs (except DLPFC) had a significant increase in explained variance. This is not apparent in the figure, not whether these statistic survive correction. Please indicate this using symbols above the pairs of whisker plots. Presumably you would need to apply correction for n ROIs (and optionally also use symbols for uncorrected significances).

4) Fig 4E: again, could you please indicate (using symbols) statistical significance levels of eye position reconstructions for the different ROIs?

Minor:

1) L88: “Here, we combined areas into single regions of interest (ROI) with approximately the same number of voxels”. Move this sentence below the next one. And: without reading methods, it is not clear what was done: Did you sample voxels randomly in each area to make voxel-matched ROIs? Please Explain.

2) L108ff: it is unnecessarily unclear when and how the two contrast levels were applied: you had 2 contrasts. One “after saccade”, one “during fixation”. Clarify this sentence to describe how (abrupt change or smooth transition) and when (after 333ms?) you changed contrast settings from “after saccade” to “during fixation”. You can clarify this (by some simple rewording) without using (much) more space, but it will save readers diving into the methods or waiting for the following 2 paragraphs where this becomes clearer.

Signed review: Andreas Bartels

Reviewer #2 (Remarks to the Author):

In species that move their eyes, establishing spatial representations requires integrating the visual signals impinging onto the retina with information about the direction of gaze. At the neural level, a way in which this process is believed to take place is via modulation of visual responses by eye position. Such a modulation has been observed at multiple stages of the visual hierarchy and described as a multiplicative gain field. The study by Fabius et al build on this literature to examine gain field characteristics in humans. The authors developed an encoding model that combines voxel-wise retinotopic prediction of activity based on visual stimulation together with population eye-position dependent gain fields. They report that inclusion of gain fields in the model enables both better prediction of measured activity in all visual areas and reconstruction of eye movement. Furthermore, they observe a specific pattern in the overall organization of gain fields.

I find the approach interesting, and I appreciate the sophisticated modeling of the data. However, I don't find the results particularly striking given what is already known in the literature. Specifically, the notion that activity could be better modeled by an encoder that also incorporates eye position is somewhat expected given the large body of preceding work in the field. As the authors themselves note in their review of the literature, the presence of gain fields at various stages of the visual pathways has been well established by neurophysiological recordings in macaques and already confirmed by imaging studies in humans. Thus, while the technique for estimating eye modulation can be useful, the conclusion that gain fields are common in the visual system is not novel.

The manuscript also argues that the proposed technique allows accurate reconstruction of eye position over time, but this claim is not well supported. Figure 5 which is dedicated to this issue only reports one example trace and correlation values. It is difficult to understand how good the model really is: the axes in panel B do not have units, and the y correlation in panel C does not seem high, certainly not enough for concluding that the reconstruction is accurate. More importantly, a more thorough assessments than just correlation would be needed to claim accurate reconstruction of eye movements.

Given the comments above, the main novel observation of this manuscript seems to be restricted to the organization of gain field in relation to eccentricity. This is an interesting finding, but one that is for me difficult to disentangle from the many technical choices that have been made in the development of the model. As I understand it, the encoder heavily relies on ad hoc modeling of the visual field and its effects at different times relative to the occurrence of saccades. This seems recursive, raising the question of whether the structure of gain fields could have been affected by these choices or by inadvertently introducing biases in the way the visual scene is processed or at other stages of the model.

General reply

We thank the reviewers for their comments. We have incorporated reviewers' suggestion, added figures/analyses to corroborate our answers and modified relevant sections in the main manuscript. We believe that the manuscript has improved substantially from its original version. In the following we address the comments and the concerns raised.

Reviewer comments are reported in italics, text in black; our rebuttal in roman font, text in black. Updated sections in the main manuscript and supplementary materials are in light grey text, roman font. In the main manuscript file, updated sections are highlighted in green.

While working on incorporating the reviewers' suggestions we noticed a small error in our code for processing the reconstructed eye position. When computing the correlation between the reconstructed and actual eye position, we accidentally used the actual x coordinate for both the x and y coordinates. In the previous version of the manuscript, we reported the correct correlation in figure 4D, but figure 4C reported the wrong values. We have now corrected and updated the figure and the analysis.

After correcting this error, the correlations between the actual and reconstructed y coordinates are slightly higher. Moreover, the estimated lag between reconstruction and actual eye position decreased from 2 TRs to 1 TR – as estimated with a weighted average from the maximum cross correlations between actual and reconstructed x and y coordinates. Correcting for this error did not lead to any changes in the interpretation of the data. All statistics keep the same significance, although numerically slightly different. In short, we noticed and corrected a small error in the analysis scripts of the reconstructed eye positions, which lead to numerical changes in the results section, but which did not change the interpretation of the statistics and conclusions.

Reviewer #1

The authors conducted three human fMRI experiments with and without saccadic eye movements over moving or static-flickering bar-graph stimuli in order to systematically model the influence of eye positions on the fMRI responses to visual bar stimuli. Using extensive modelling of population receptive fields (pRFs) as well as population eye position gain fields (pEGFs) the authors show that eye-positions indeed modulate neural responses in a systematic way, and that pEGFs centers tend to anti-correlate with eccentricity of the pRFs. Overall, this is a well conducted, well presented, highly readable, timely, and important paper that will receive considerable attention from both, visual, motor, and theoretical neuroscientists. The methods are sound and well documented.

While I recommend publication, I have several major and minor points regarding some aspects of modelling, presentation of statistics, and concepts in the introduction.

Major

Introduction

1) *I like that you try to introduce the reader from scratch to the topic. However, it is important to not conflate concepts and keep things simple. Your art example is beautiful, but why then refer to another piece of art that is not shown (Pollock)? L32ff: "With only the retinotopic input at our disposal, we would quickly lose track of all the stars and other elements. As a result, "La nuit étoilée" would become barely indistinguishable from Jackson Pollock's "Night mist". This statement lacks conceptual motivation, for two reasons: First, you introduced the eye gain field problem in terms of attribution of movement (self vs other), not in terms of improving recognition of images. Second, also a brief fixation*

(with no eye movements) will allow readers to see and differentiate between two paintings. Suggestion: cut L32ff.

We thank Dr. Bartels for his review and kind words. We agree with the two points about the reference to the second piece of art and removed it.

Results

2) *L134 ff: the approach of including peripheral visual cues outside the screen is interesting, but requires more detailed description and statistics in order demonstrate its usefulness (or lack thereof) (both would add important knowledge to the field)*

2a) *You go to great lengths to model and maximize the variance explained by not only modelling the on-screen stimulus, but also the static peripheral cues (screen border etc). Fig 2 quantifies several parameters, the main text even more. However, in the end, the reader is left uncertain about the benefits of this is approach: the model fits best when “during fixation” the peripheral contrast is 0. The effects “after saccade” are overall small. This leaves me wonder: does inclusion of the periphery make a statistically significant difference, and how much, compared to leaving it out completely? How large is this effect across cortical regions? Please provide a plot of optimal modelling peripheral contrast vs not at all, comparing the respective R square, with statistical tests (F-tests?), across visual regions (analogue to Fig. 3B). This would be useful information to inform future experiments.*

We have added the suggested analysis and figure to the Supplementary Information: Supplementary Information → Statistic output → 1. Linear mixed-effects model of change in R2 between the pRF model not optimized for stimulus configuration and the pRF model optimized for stimulus configuration (pRF-only model), per ROI.

To summarize the results, when we added the peripheral elements with the optimal configuration (pRF-only model)**, there was a significant increase of around 0.035 variance explained in V1, V2, IPS0-1, IPS2-3-4 and TO1/TO2. The increase in additional variance explained was not significant and around 0.015 in V3, V4, V3AB and LO1/LO2. In our control area, located in the DLPFC, the additional variance explained was 0.009 (and not significant). These increases in variance explained are medians across voxels and subjects.

While these increases in variance explained are indeed small, we believe it is beneficial to keep the description of how we model static peripheral cues in the manuscript. If we were to remove it altogether, a reader might be left wondering whether these cues could account to some degree for the observed results. Whereas, by including it, we account the amount of variability captured by these cues and decrease the likelihood that they might contribute to the observed effects.

**Our decision of which parameters were optimal were based on the responses in V1. This was made explicit in the main text, but not in the legend of Figure 2D, so we have now added this to its caption.

New supplementary figure 2 and stats reported in the supplementary information:

Figure S1. Change in the variance explained (R^2) of BOLD responses in the eye-movement task by the pRF-only model (optimized for stimulus configuration, right boxes) compared to the pRF model not optimized for stimulus configuration (left boxes). For each pair of boxes, the left box represents the R^2 of the model that did not include the peripheral elements, the right box represents the R^2 of the model where the peripheral elements were modelled with full contrast for a period of 500 ms after saccade offset (pRF-only model). Grey lines and points represent single participants. R^2 is computed as the median over all voxels of the most visually responsive voxels (i.e. the same as for Figure 3B). Asterisks indicate a significant difference (see all test results in Supplementary Information – Statistics Output – 1.). Overall, the pRF-only model yielded an increase in R^2 of approximately 0.035 in V1, V2, IPS0-1, IPS2-3-4 and TO1-2, compared to the pRF model not optimized for stimulus configuration.

1. Linear mixed-effects model of change in R^2 between the pRF model not optimized for stimulus configuration and the pRF model optimized for stimulus configuration (pRF-only model), per ROI

Command:

```
anova(md10, md11)
```

Models:

```
md10: dR2 ~ 0 + (1 | subj)
```

```
md11: dR2 ~ roi + (1 | subj)
```

	npar	AIC	BIC	logLik	deviance	Chisq	Df	Pr(>Chisq)
md10	2	-510.11	-504.71	257.06	-514.11			
md11	12	-552.25	-519.85	288.13	-576.25	62.14	10	1.423e-09 ***

Command:

```
anova(md11)
```

Type III Analysis of Variance Table with Satterthwaite's method

	Sum Sq	Mean Sq	NumDF	DenDF	F value	Pr(>F)
roi	0.016269	0.0018077	9	90	7.1565	9.188e-08 ***

Confidence intervals of the changes in R^2 per ROI. Estimates are derived by bootstrapping md11 100000 times using the “bootMer” and “confint” functions from the “boot” package in R. Estimates are corrected for multiple comparisons using Bonferroni’s correction, by adjusting the limits of the confidence interval by the number of comparisons (= 10). I.e. the limits of a two-sided 95%-confidence

interval become the $(2.5/10) = 0.25^{\text{th}}$ percentile and $(100-2.5/10) = 99.75^{\text{th}}$ percentile. Significance is determined by whether or not 0 is included in the confidence interval.

ROI	median	q025	q975	q025 corrected	q975 corrected	significant [†]
V1	0.0391	0.0230	0.0550	0.0162	0.0630	*
V2	0.0300	0.0139	0.0460	0.0068	0.0530	*
V3	0.0158	-0.0003	0.0320	-0.0072	0.0390	
V4	0.0142	-0.0019	0.0300	-0.0089	0.0370	
V3AB	0.0159	-0.0001	0.0320	-0.0071	0.0390	
IPS0-1	0.0348	0.0188	0.0510	0.0120	0.0580	*
IPS2-3-4	0.0355	0.0194	0.0520	0.0126	0.0590	*
LO1/LO2	0.0163	0.0002	0.0320	-0.0064	0.0390	°
TO1/TO2	0.0360	0.0200	0.0520	0.0129	0.0590	*
DLPFC	0.0009	-0.0151	0.0170	-0.0220	0.0240	

[†]Significance classification:
 * = 0 outside Bonferroni-corrected 95%-confidence interval
 ° = 0 outside uncorrected 95%-confidence interval

2b) L134ff: “The median variance explained”: it is unclear what this refers to: the best-fitting model, or the additional variance explained by including the peripheral static field? Please clarify.

This refers to the variance explained by the pRF model in combination with the optimized stimulus configuration (pRF-only model). We have added this clarification to the manuscript.

L148-157: The optimal configuration of the peripheral elements incorporated into the stimulus model consisted of post-saccadic contrast increases (for 500 ms, from 0 to 100%) and no contrast during periods of fixation. The optimal configuration was set based on the increase in R^2 in V1 voxels. This stimulus configuration led to a significant increase in R^2 of approximately 0.035 in V1, V2, IPS0-1, IPS2-3-4 and TO1-2, but not in the other areas (Figure S2; Supplementary Information – Statistics Output – 1).

To summarize, in line with single cell recordings²⁷, the static peripheral elements act as a visual stimulus when they are brought into a pRF after a saccade (Fig. 2D). The median variance explained by the pRF model in combination with the optimized stimulus configuration (pRF-only model) varied between 0.30 (V1, V2, V3) and 0.14 (LO1-2) across all visual areas. Thus, pRFs can be used to capture visual responses in active eye-movement tasks when the visual input is modelled adequately.

3) Fig 3B: if I read the sup mat “statistics output” correctly, then all ROIs (except DLPFC) had a significant increase in explained variance. This is not apparent in the figure, not whether these statistics survive correction. Please indicate this using symbols above the pairs of whisker plots. Presumably you would need to apply correction for n ROIs (and optionally also use symbols for uncorrected significances).

We have updated Figure 3B as suggested, using an asterisk above the pairs of whisker plots (see below). Moreover, we have updated the table with the statistics output in the Supplementary Information which now shows both the corrected (as it showed previously) and uncorrected bootstrapped 95%-confidence intervals (CI) (see Supplementary Material – Statistics Output – 2). In addition, we have changed the “significance” column to display an asterisk (*) when zero is outside the corrected CI and a degree symbol (°) when zero is outside the uncorrected but inside the corrected CI. In the manuscript we only refer to a significant difference when zero is outside the corrected CI.

Main manuscript, updated Figure 3B, now shows an asterisk above the pairs of box plots for significant comparisons.

Statistics reported in the supplementary material (Supplementary Material – Statistics Output – 2). Confidence intervals of the changes in R^2 per ROI. Estimates are derived by bootstrapping mdl1 100000 times using the “bootMer” and “confint” functions from the “boot” package in R. Estimates are corrected for multiple comparisons using Bonferroni’s correction, by adjusting the limits of the confidence interval by the number of comparisons (= 10). I.e. the limits of a two-sided 95%-confidence interval become the $(2.5/10) = 0.25^{\text{th}}$ percentile and $(100-2.5/10) = 99.75^{\text{th}}$ percentile. Significance is determined by whether or not 0 is included in the confidence interval.

2. Linear mixed-effects model of change in R^2_{adj} per ROI as a result of the addition of the pEGF to the pRF-only model

```
Command:
anova mdl0, mdl1)

Models:
mdl0: dR2adj ~ 0 + (1 | subj)
mdl1: dR2adj ~ roi + (1 | subj)

      npar      AIC      BIC logLik deviance  Chisq Df Pr(>Chisq)
mdl0    2 -517.20 -511.79 260.60 -521.20
mdl1   12 -673.05 -640.64 348.52 -697.05 175.85 10 < 2.2e-16 ***

Command:
anova(mdl1)

Type III Analysis of Variance Table with Satterthwaite's method
      Sum Sq  Mean Sq NumDF DenDF F value  Pr(>F)
roi 0.026515 0.0029461    9    90  30.77 < 2.2e-16 ***
```

Confidence intervals of the changes in R^2_{adj} per ROI and differences in changes between ROIs. Estimates are derived by bootstrapping mdl1 100000 times.

ROI 1	ROI 2	median	q025	q975	q025 corrected	q975 corrected	significant [†]
V1	-	0.0593	0.0523	0.0664	0.0474	0.0713	*
V2	-	0.0526	0.0456	0.0596	0.0407	0.0643	*
V3	-	0.0545	0.0476	0.0615	0.0426	0.0665	*
V4	-	0.0344	0.0274	0.0414	0.0226	0.0463	*
V3AB	-	0.0393	0.0323	0.0462	0.0272	0.0512	*
IPS0-1	-	0.0271	0.0200	0.0341	0.0153	0.0386	*

IPS2-3-4	-	0.0251	0.0181	0.0322	0.0135	0.0369	*
LO1/LO2	-	0.0370	0.0300	0.0440	0.0250	0.0489	*
TO1/TO2	-	0.0348	0.0278	0.0418	0.0229	0.0468	*
DLPFC	-	0.0037	-0.0033	0.0107	-0.0081	0.0155	
V1	V2	0.0068	-0.0014	0.0150	-0.0072	0.0204	
V1	V3	0.0048	-0.0033	0.0130	-0.0092	0.0189	
V1	V4	0.0249	0.0167	0.0331	0.0110	0.0388	*
V1	V3AB	0.0201	0.0119	0.0283	0.0064	0.0339	*
V1	IPS0-1	0.0323	0.0240	0.0404	0.0186	0.0463	*
V1	IPS2-3-4	0.0342	0.0260	0.0424	0.0204	0.0480	*
V1	LO1/LO2	0.0223	0.0141	0.0305	0.0086	0.0361	*
V1	TO1/TO2	0.0245	0.0163	0.0327	0.0103	0.0384	*
V1	DLPFC	0.0556	0.0474	0.0638	0.0419	0.0697	*
V2	V3	-0.0020	-0.0101	0.0062	-0.0157	0.0120	
V2	V4	0.0182	0.0100	0.0264	0.0043	0.0320	*
V2	V3AB	0.0133	0.0051	0.0215	-0.0007	0.0270	o
V2	IPS0-1	0.0255	0.0173	0.0337	0.0117	0.0393	*
V2	IPS2-3-4	0.0274	0.0192	0.0356	0.0133	0.0414	*
V2	LO1/LO2	0.0155	0.0074	0.0237	0.0017	0.0295	*
V2	TO1/TO2	0.0178	0.0096	0.0259	0.0037	0.0315	*
V2	DLPFC	0.0488	0.0406	0.0571	0.0350	0.0624	*
V3	V4	0.0201	0.0120	0.0283	0.0065	0.0341	*
V3	V3AB	0.0153	0.0071	0.0234	0.0014	0.0293	*
V3	IPS0-1	0.0274	0.0193	0.0357	0.0137	0.0413	*
V3	IPS2-3-4	0.0294	0.0212	0.0375	0.0157	0.0431	*
V3	LO1/LO2	0.0175	0.0093	0.0257	0.0038	0.0314	*
V3	TO1/TO2	0.0197	0.0115	0.0279	0.0059	0.0336	*
V3	DLPFC	0.0508	0.0427	0.0590	0.0370	0.0648	*
V4	V3AB	-0.0049	-0.0130	0.0033	-0.0187	0.0089	
V4	IPS0-1	0.0073	-0.0009	0.0155	-0.0066	0.0210	
V4	IPS2-3-4	0.0093	0.0010	0.0174	-0.0047	0.0232	o
V4	LO1/LO2	-0.0026	-0.0108	0.0055	-0.0164	0.0111	
V4	TO1/TO2	-0.0004	-0.0086	0.0077	-0.0141	0.0135	
V4	DLPFC	0.0307	0.0226	0.0389	0.0168	0.0444	*
V3AB	IPS0-1	0.0122	0.0040	0.0204	-0.0017	0.0263	o
V3AB	IPS2-3-4	0.0141	0.0060	0.0223	0.0002	0.0280	*
V3AB	LO1/LO2	0.0022	-0.0060	0.0104	-0.0115	0.0163	
V3AB	TO1/TO2	0.0045	-0.0037	0.0127	-0.0094	0.0186	
V3AB	DLPFC	0.0355	0.0274	0.0437	0.0216	0.0493	*
IPS0-1	IPS2-3-4	0.0019	-0.0062	0.0101	-0.0121	0.0159	
IPS0-1	LO1/LO2	-0.0100	-0.0181	-0.0018	-0.0237	0.0043	o
IPS0-1	TO1/TO2	-0.0077	-0.0160	0.0004	-0.0218	0.0061	
IPS0-1	DLPFC	0.0234	0.0152	0.0315	0.0093	0.0369	*
IPS2-3-4	LO1/LO2	-0.0119	-0.0200	-0.0037	-0.0258	0.0019	o
IPS2-3-4	TO1/TO2	-0.0097	-0.0178	-0.0015	-0.0237	0.0043	o
IPS2-3-4	DLPFC	0.0214	0.0133	0.0296	0.0077	0.0352	*
LO1/LO2	TO1/TO2	0.0022	-0.0059	0.0104	-0.0116	0.0159	
LO1/LO2	DLPFC	0.0333	0.0252	0.0415	0.0195	0.0474	*
TO1/TO2	DLPFC	0.0311	0.0229	0.0393	0.0173	0.0452	*

[†]Significance classification:

* = 0 outside Bonferroni-corrected 95%-confidence interval

o = 0 outside uncorrected 95%-confidence interval

4) Fig 4E: again, could you please indicate (using symbols) statistical significance levels of eye position reconstructions for the different ROIs?

We have added red asterisks to indicate statistical significance to this panel and updated the caption accordingly (see new Figure 4E in the main manuscript, also reported below).

Main manuscript, Figure 4E, now shows a red asterisks to indicate the group average is statistically significant from zero

E. Correlations between actual eye position and reconstructed positions from single visual ROIs and a control area in the dorsolateral prefrontal cortex (DLPFC). The horizontal and vertical components were combined into a single correlation estimate (the Fisher transformed average of the two components) to compare the ROIs with each other. Correlations per ROI were estimated by bootstrapping a fixed number of voxels of each ROI per participant. This ensures that differences in reconstruction quality are not due to differences in size of the ROIs. Red asterisks above the boxplot indicate that the group median is significantly different from zero (i.e. that the zero is outside the Bonferroni-corrected bootstrapped 95%-confidence interval).

Minor

1) L88: *“Here, we combined areas into single regions of interest (ROI) with approximately the same number of voxels”.* Move this sentence below the next one. And: *without reading methods, it is not clear what was done: Did you sample voxels randomly in each area to make voxel-matched ROIs? Please Explain.*

We have modified the order of sentences as suggested. We combined ROIs to account for differences in size between ROIs in the reconstruction of eye trajectory. We randomly sampled voxels in each area. Please see the following changes in the main text:

L255-259: To evaluate the quality of the pEGFs across the visual hierarchy, we assessed if and to what extent eye position can be reconstructed from single ROIs. We computed correlation estimates between reconstructed and actual eye positions by bootstrapping a fixed number of voxels (≥ 150) per ROI. This ensures that potential differences in reconstruction quality are not due to the number of voxels used for the reconstruction.

2) L108ff: *it is unnecessarily unclear when and how the two contrast levels were applied: you had 2 contrasts. One “after saccade”, one “during fixation”.* Clarify this sentence to describe how (abrupt change or smooth transition) and when (after 333ms?) you changed contrast settings from “after saccade” to “during fixation”. You can clarify this (by some simple rewording) without using (much) more space, but it will save readers diving into the methods or waiting for the following 2 paragraphs where this becomes clearer.

We have reworded this sentence:

L116-121: First, we varied the contrast of the peripheral elements from 0 to 100% with respect to the contrast of the bars. There were two moments of contrast increase of these elements: one after a saccade (for 333 ms, i.e. two frames at 6 Hz) and one during fixation (all the other frames). The contrast

change was implemented instantaneously, without continuous transition (i.e. contrast incrementally increased between frames), because this would not be possible for the two frames of contrast increase after a saccade.

Signed review: Andreas Bartels

Reviewer #2

In species that move their eyes, establishing spatial representations requires integrating the visual signals impinging onto the retina with information about the direction of gaze. At the neural level, a way in which this process is believed to take place is via modulation of visual responses by eye position. Such a modulation has been observed at multiple stages of the visual hierarchy and described as a multiplicative gain field. The study by Fabius et al. build on this literature to examine gain field characteristics in humans. The authors developed an encoding model that combines voxel-wise retinotopic prediction of activity based on visual stimulation together with population eye-position dependent gain fields. They report that inclusion of gain fields in the model enables both better prediction of measured activity in all visual areas and reconstruction of eye movement. Furthermore, they observe a specific pattern in the overall organization of gain fields.

1) *I find the approach interesting, and I appreciate the sophisticated modelling of the data. However, I don't find the results particularly striking given what is already known in the literature. Specifically, the notion that activity could be better modelled by an encoder that also incorporates eye position is somewhat expected given the large body of preceding work in the field. As the authors themselves note in their review of the literature, the presence of gain fields at various stages of the visual pathways has been well established by neurophysiological recordings in macaques and already confirmed by imaging studies in humans. Thus, while the technique for estimating eye modulation can be useful, the conclusion that gain fields are common in the visual system is not novel.*

We thank the reviewer for finding interest in our work and modelling approach. In the following we argue in defense of the novelty of our findings. We agree with the reviewer that eye-position dependent gain modulation has been reported along several stages of the visual hierarchy. However, our investigation provides several new insights that we believe are of general interest.

1. Current studies on human gain fields provide proof of principle for the existence of the mechanism in humans. However, they do not provide information about their properties. The systematic characterization of gain field properties allows to go beyond the proof of principle and start asking questions about their distributional features (e.g. are there biases for a particular viewing direction?); the nature of the mechanism; and potential topographical features. In the current investigation, after describing how we characterize gain fields in a dynamic setting (see point 3 below), we focus in particular on the topographic organization of gain fields. The idea of a topographic organization in gain field position has been put forward in the discussion of Andersen et al., 1985 and has been a point of debate ever since, with some studies describing hints of topography in some cortical areas (Galletti et al., 1989, *J. Neurosci*; Durand et al., 2010, *Neuron*; Guo et al., 1997, *Neuroreport*) and others observing no systematicities at the population level (Bremmer et al., 1997, *J. Neurophysiol.*; Sakata et al., 1980, *J. Neurophysiol*; Galletti et al., 1995, *Eur. J. Neurosci.*). Please note that these references are listed in the main manuscript.
2. While gain fields have been observed in human early visual cortex, they have not been reported previously in human parietal cortex (areas IPS1-IPS5).
3. In human imaging experiments (and most monkey neurophysiology studies), the gain field mechanism is studied under static conditions: the participant fixates while being presented with a visual stimulus. We introduce a novel approach by probing the gain field mechanism in an active setting, with the participant moving their eyes along given trajectories while being presented with visual stimuli. We cannot take for granted that a mechanism studied in a passive setting would transfer to an active setting following the same rules. For example, the involvement of the motor system or refference, could in principle lead to different results.

4. Our approach allows to exclude potentially spurious sources of gain modulation due to residual retinotopic visual stimulation at the borders of the visual field of view, or intrinsically associated with performing eye-movements. While this potential confound can be controlled with relative ease in neurophysiology experiments with monkeys, in human imaging experiments it can be more complex. In the past, this concern has been accounted for in human imaging studies in several ways, for example by excluding voxels that might be affected by residual retinotopic visual stimulation or by attempts to remove potential retinotopic stimulation outside the field of view (Fischer et al., 2012, *Neuron*; Merriam et al., 2013, *J.Neurosci*; Strappini et al., 2015, *Brain Struct. And Func.*). The novelty of our approach lies in a careful control of these confounds while participants actively perform eye-movements across the visual field, taking into account both the individual retinotopy and population receptive fields (pRF) characteristics (e.g. size by eccentricity relationship, potential asymmetries in pRF distribution or coverage), as well as the potential retinotopic activity elicited by the border of the visual stimulation.

We have now included a summary and discussion of these points in the introduction / discussion of the revised manuscript.

Introduction

L52-74: A systematic characterization of gain-field properties allows us to ask questions about the distributional features of EGFs (e.g. are there biases for a particular viewing direction?) and its extension to human parietal cortex. Moreover, it offers the possibility to probe the existence of a potential underlying topographical organization for EGFs. This idea has been originally put forward in the discussion of Andersen and colleagues (1985), and has been a point of debate ever since, with some studies describing hints of topography in some cortical areas^{12,17,18} and others observing no systematicities at the population level^{8,9,11,13}.

Here, we present an encoding model that characterizes visual responses and EGFs at the level of single voxels in human 7T functional magnetic resonance imaging (fMRI) data. EGFs have been generally studied under static conditions both in human and non-human primates, with participants keeping stable fixation while being presented with a visual stimulus²⁰⁻²². However, we cannot a priori assume that a mechanism studied in a passive setting would transfer in an active setting following the same principles. We introduce a novel approach by probing EGFs in an active setting, with participants moving their eyes along given trajectories while being presented with high contrast visual stimuli.

The model follows a parsimonious approach and is built on first principles: the retinotopic organization of visual cortex and its response to contrast. First, we show that the population receptive field model (pRF)²³ can capture visual responses elicited by eye-movements that bring a stimulus into a pRF, capturing contrast-based responses from visual cortex in an active setting. Second, building upon the pRF responses, we estimate population eye-position dependent gain fields (pEGF) as two dimensional gaussians, which we validate by reconstructing eye positions from an independent dataset. Third, our encoding model allows for the exploration of previously unknown, large-scale systematicities in the distributions of EGF parameters at the population level. In early visual areas we observed that pEGF centers follow a topographic organization along the eccentricity of pRFs.

Discussion

L337-341: The systematic characterization of pEGF properties was possible throughout the human visual system, also including the human parietal cortex. We examined the distribution of pEGF parameters at the population level and discovered a large-scale organization of pEGF centers: following the gradient of pRF eccentricity, pEGF centers shift from contralateral (for central pRFs) to ipsilateral (for peripheral pRFs).

L390-398: It is important to note that the majority of these studies did not examine a relationship between RF location and EGF location, but rather examined the distribution of EGFs in their recorded sample of neurons. It is possible that an explicit relationship is only visible when EGF centers are

expressed as a function of RF eccentricity. Otherwise, a bias in EGF centers would only be visible if the RFs of neurons in the recorded sample would be clustered around a single eccentricity. Moreover, both contra- and ipsilateral biases in EGF centers have been observed. Our data reconcile these findings with the demonstration of a topographic gradient of pEGF centers from contra- to ipsilateral following pRF eccentricity. Thus, corroborating the original idea of a topographic organization in gain-field position initially put forward in the discussion of Andersen and colleagues (1985).

2) *The manuscript also argues that the proposed technique allows accurate reconstruction of eye position over time, but this claim is not well supported. Figure 4 which is dedicated to this issue only reports one example trace and correlation values. It is difficult to understand how good the model really is: the axes in panel B do not have units, and the y correlation in panel C does not seem high, certainly not enough for concluding that the reconstruction is accurate. More importantly, a more thorough assessments than just correlation would be needed to claim accurate reconstruction of eye movements.*

In Figure 4 we report the median reconstructed trajectory in x and y coordinates across participants, where the grey area around the line represents the inter-quartile range.

To represent the strength of the observed correlations more transparently, we now show the units of the axes in panel 4B (observed/actual and estimated/reconstructed eye positions for an individual participant in degrees of visual angle). Figure 4C now shows the distribution of observed Pearson correlations across participants. The coefficient of correlation ranges between -1 and 1, with 1 indicating perfect association between the variables, 0 indicating no association between the variables (dotted line in figure 4C) and -1 indicating perfect negative correlation between the variables. Now, the y-axis spans this full range. We only observed correlations above zero, with a median correlation for the y coordinate of 0.64 and a median correlation for the x coordinate of 0.77. According to Cohen (Cohen, 1988, 2013), a correlation value of 0.4 indicates a “medium” strength of association, whereas a value of 0.7 indicates a “large” or “strong” association (also reported in Hemphill, 2003).

Main manuscript, new figures 4B and 4C, now with unit specification in 4B and the y-axis spanning the full range of possible correlation values in 4C.

In Figure R1, we report the same figure as in Figure 4C, using several correlation measures and the average Euclidian distance between the expected and reconstructed eye-movement trajectory in degrees of visual angle, per individual participant. This measure represents the difference between the expected and reconstructed trajectory in relevant units, degrees of visual angle, thus going beyond simple correlation. These measures are suggestive of a robust association between the expected and reconstructed eye-movement trajectories, indicating that our model accurately captures visuo-motor behavior. It is important to note that these measures were obtained in a completely independent task and starting from the results of population gain-fields obtained at an individual subject level.

In Figures R2 and R3 (reported below), we show scatter plots similar to the example reported in Figure 4B for all participants (Figure R2, x coordinate, and Figure R3, y coordinate).

To conclude, our correlation and distance measures indicate an accurate reconstruction of eye position based on our gain field model.

Cohen, J., 1988, 2013. Statistical power analysis for the behavioral sciences. Routledge.

Hemphill, J.F., 2003. Interpreting the magnitudes of correlation coefficients.

Figure R1. Similarity measurements (left panel) and distance measurements (right panel). Similarity measures: Kendall rank correlation, Pearson correlation and Spearman rank correlation. Distance measures: Euclidian distance between the expected and the reconstructed eye-movement position (average over time for each participant).

Figure R2. Reconstructed against actual eye position for all participants – horizontal component. The title above each panel shows the Pearson's correlation coefficient.

Figure R3. Reconstructed against actual eye position for all participants – vertical component. The title above each panel shows the Pearson's correlation coefficient

3) Given the comments above, the main novel observation of this manuscript seems to be restricted to the organization of gain field in relation to eccentricity. This is an interesting finding, but one that is for me difficult to disentangle from the many technical choices that have been made in the development of the model. As I understand it, the encoder heavily relies on ad hoc modeling of the visual field and its effects at different times relative to the occurrence of saccades. This seems recursive, raising the question of whether the structure of gain fields could have been affected by these choices

or by inadvertently introducing biases in the way the visual scene is processed or at other stages of the model.

We agree with the reviewer that the large-scale organization of gain fields represents a novel aspect presented in the manuscript (although not the only one, please refer to the points covered in the reply to reviewer 2, question 1). In the following we comment on our modelling approach, with a focus on the problem of recursiveness and on the technical choices we made.

Recursiveness vs. generalization

The logic behind this work follows a parsimonious approach and is based on first principles. We assumed that any observation we made was spurious unless we could prove otherwise. Modelling the expected responses for each voxel (each population receptive field) for different contrasts and time relative to the saccade is a consequence of this approach. In the following, we use the word ‘training’, with reference to fitting the parameters of the pEGF per voxel, and the word with ‘testing’ with reference to reconstructing eye positions from an independent dataset.

We needed to exclude potentially spurious sources of variability due to residual retinotopic visual stimulation and the borders of the visual field of view. Here we based our strategy on the known underlying retinotopic organization of visual cortex and its response to contrast.

First, we show that population receptive fields (measured on an independent dataset) capture the variability of visual responses induced by eye-movements, including peripheral elements. The model captures about 30% of the variability in early visual cortex (V1-V2-V3) and about 17% in the parietal region (see Figure 3B, pRF-only model results). Importantly, we did not know a priori the effect of contrast-based response after an eye-movement within the scanner. Thus, we opted for testing a bank of contrasts and times with respect of saccade onset (see figure 2), to capture the maximum of variability linked to this modulation.

Second, we trained our model on one particular combination of trajectories and visual stimulation. We tested the model on a completely independent fMRI dataset, in which the participants were required to follow an independent trajectory, and hence resulting in a different visual stimulation. This strategy allowed to test the generalizability of our modelling results.

The inclusion of peripheral elements did not account for a considerable amount of variability (~3.5%, see response to reviewer 1, question 2 and Supplementary Figure 2). Nevertheless, we included this contribution in the model for an important reason. If we were to remove the contribution of peripheral elements, we might be left wondering whether they could partially account for the observed reconstruction quality or the large-scale organization of gain field parameters. Whereas, by including the contribution of peripheral elements (as well as contrast increase time relative time relative to saccade onset), we can account for the amount of variability captured by these cues, hence reducing the likelihood that our model would wrongly attribute this variability to a gain field modulation, leading to a spurious interpretation of the results.

Importantly, the approach would be recursive if we were training and testing our model on the same data or on an identical version of the experiment ran on a second session (e.g. with the same contrast stimulation). In that case there could have been a problem of overfitting. However, we trained and optimized the model on one particular combination of trajectories and visual stimulation and tested it on a completely independent fMRI dataset, with orthogonal trajectories. If our results were due to overfitting, then the model would not generalize to the independent fMRI dataset.

Technical choices

Once established that our modelling results were not due to residual retinotopic visual stimulation or the borders of the visual field of view, it was necessary to exclude the possibility that our results were driven by biases introduced by the modelling strategy per se. For example, it could be that, starting from uniform distributions of gain field parameters, the modelling and fitting procedure would introduce distributional features that are not present at the input stage.

To account for this possibility, we worked extensively on model simulations. Following this approach, we observed that some features present in the data could be attributed to our modelling strategy (e.g. the central distributional peak and two peaks at the tails (see figure 5C). However, the shift between pRFs on the left or right hemifield was not present in our simulations. For this reason, we conclude that the shift and the associated link with eccentricity represents a true physiological property of gain fields.

This type of approach comes inevitably with a level of technicality, as we needed to formalize the model, the starting distributions and the readout strategy. However, all these choices allowed a careful assessment of the genuine or spurious nature of our findings and were justified according to parsimony and first principles: the known underlying retinotopic organization of visual cortex and its response to contrast.

In conclusion, starting from a skeptical position (assuming that any observation we made was spurious, unless we could prove otherwise) and using a parsimonious approach we can conclude that: i) our results are not affected by the problem of recursiveness, otherwise the results would not have generalized to a completely independent fMRI dataset with orthogonal trajectories; and ii) the results we interpret are not driven by biases introduced by the modelling strategy per se.

Changes

L345-360 Previous fMRI studies that studied eye position responses or modulation of visual responses by eye position have been performed in complete darkness³⁴, with the use of motion-defined instead of luminance defined stimuli and covering the inside of the scanner bore with black felt²⁰ or only examining voxels responsive to a central region of the visual field^{21,22}. Here, we took a different approach, implemented a design where participants were actively performing eye-movements in the scanner and factored peripheral visual stimulation into the retinotopic representation of our stimulus. We examined how the peripheral, static elements in the visual field should be incorporated into the representation. With this approach, we were able to adopt a relatively naturalistic eye-movement task for the participant, without the need to fixate at a single point for several minutes.

Next, it was necessary to exclude the possibility that our results were driven by biases introduced by our modelling strategy per se. For example, it could be that starting from uniform distributions of pEGF parameters, the modelling and fitting procedure would introduce distributional features not present at the input stage (Fig. 5C). Moreover, it is possible that our representation of the retinotopic input was either incomplete (i.e., some elements providing strong visual responses were omitted) or included elements that did not provide strong stimulation. To account for these possible confounds, we worked extensively on model simulations.

L367-375 Together, these simulations indicate that accurate reconstruction of eye position over time is tightly coupled with the existence of eye-position dependent gain modulation of visual responses and is not a result of unaccounted retinotopic stimulation.

This type of approach comes inevitably with a level of technicality, as we needed to formalize the model, the starting distributions, and the readout strategy. However, all these choices allowed for a careful disentanglement of genuine and spurious findings and were justified according to parsimony and first principles: the known underlying retinotopic organization of visual cortex and its response to contrast.

REVIEWER COMMENTS

Reviewer #1 (Remarks to the Author):

The authors spent a good effort to address all my comments in full. I congratulate them on a really beautiful study and highly readable and accessible manuscript. I have no further comments.

Reviewer #2 (Remarks to the Author):

I find this article improved. It now does a better job in explaining its significance as well as the authors' confidence in concluding that response modulations originate from oculomotor gains.

I do not have further major comments regarding the manuscript, but I am puzzled by the statement in the reply letter that, after correcting for the previous error in correlating two unrelated traces (predicted eye position on one axis with measured position on the other) results change only slightly. This cannot be true unless the horizontal and vertical displacements were highly correlated (or the predicting power of the model is close to zero). I must be missing something here.

Minor comments

Figure 1E is quite confusing. I think the point of the figure is to simply show that subjects followed two different eye movement trajectories. But the axes are unlabeled (I assume eye positions) and the trajectories actually look quite similar to each other.

Figure 3A. The black line with the original data is not labeled in the figure.

Figure 4A. Add "gaze position" as a label on y axis. Also the labels of the other plots can be made more explicit.

General reply

We thank the reviewers again for their compliments and comments. We were glad to see the reviewers also considered the manuscript to have improved.

We have incorporated reviewers' suggestion. Reviewer comments are reported in italics, text in black; our rebuttal in roman font, text in black.

Reviewer #1

The authors spent a good effort to address all my comments in full. I congratulate them on a really beautiful study and highly readable and accessible manuscript. I have no further comments.

Thank you.

Reviewer #2

I find this article improved. It now does a better job in explaining its significance as well as the authors' confidence in concluding that response modulations originate from oculomotor gains.

I do not have further major comments regarding the manuscript, but I am puzzled by the statement in the reply letter that, after correcting for the previous error in correlating two unrelated traces (predicted eye position on one axis with measured position on the other) results change only slightly. This cannot be true unless the horizontal and vertical displacements were highly correlated (or the predicting power of the model is close to zero). I must be missing something here.

The error spotted was only present in one analysis script. For scrutiny, we had separated each analysis as much as possible in the analysis scripts. This decision worked in our favor, as only one analysis was affected – the one presented in Figure 4B.

However, even though we did not re-use the same code across different analyses, we did use the result of the 'affected' analysis to determine the lag with which the reconstructed and actual eye positions correlate most strongly. This lag was used in all subsequent analysis. After correcting the error, the optimal lag changed from 2 to 1 TR.

In addition, the reviewer is correct that the horizontal and vertical eye positions are somewhat correlated **within** task ($r = 0.14$ within eye-movement task A; $r = -0.38$ within eye-movement task B). This was unavoidable, because we wanted to have eye movement trajectories that would result in relatively long duty cycles and would fit on the relatively small screen in the MRI bore, hence we opted for eye-movement trajectories along the screen diagonals, leading to partially correlated horizontal and vertical eye positions within a task. Please note that the correlations between x and y **within** a task do not affect the correlation of eye movement trajectories **between** the tasks ($r = -0.08$ for the horizontal, X component; $r = -0.09$ for the vertical, Y component).

Minor

Figure 1E is quite confusing. I think the point of the figure is to simply show that subjects followed two different eye movement trajectories. But the axes are unlabeled (I assume eye positions) and the trajectories actually look quite similar to each other.

We thank the reviewer for pointing this out. We have now corrected and added labels to both trajectories. To make the figure less confusing, we have also added two pointers that indicate which panels belong to which task.

Figure 3A. The black line with the original data is not labeled in the figure.

We have now labeled the black line as "Data".

Figure 4A. Add "gaze position" as a label on y axis. Also, the labels of the other plots can be made more explicit.

We have made the adjustment suggested by the reviewer. In addition, we have made the labels in Figure 4B and 4D more explicit. (Also, for aesthetic reasons, we changed the color of the asterisks in 4E from red to yellow).